# Estimating Gradients for Discrete Random Variables by Sampling without Replacement

**Wouter Kool**
University of Amsterdam
ORTEC
w.w.m.kool@uva.nl

**Herke van Hoof**
University of Amsterdam
h.c.vanhoof@uva.nl

**Max Welling**
University of Amsterdam
CIFAR
m.welling@uva.nl

## Abstract

We derive an unbiased estimator for expectations over discrete random variables based on sampling *without replacement*, which reduces variance as it avoids duplicate samples. We show that our estimator can be derived as the Rao-Blackwellization of three different estimators. Combining our estimator with RE-INFORCE, we obtain a policy gradient estimator and we reduce its variance using a built-in control variate which is obtained without additional model evaluations. The resulting estimator is closely related to other gradient estimators. Experiments with a toy problem, a categorical Variational Auto-Encoder and a structured prediction problem show that our estimator is the only estimator that is consistently among the best estimators in both high and low entropy settings.

## 1 Introduction

Put replacement in your basement! We derive the *unordered set estimator*[1]: an unbiased (gradient) estimator for expectations over discrete random variables based on (unordered sets of) samples *without replacement*. In particular, we consider the problem of estimating (the gradient of) the expectation of $f(\boldsymbol{x})$ where $\boldsymbol{x}$ has a discrete distribution $p$ over the domain $D$, i.e.

$$\mathbb{E}_{\boldsymbol{x} \sim p(\boldsymbol{x})}[f(\boldsymbol{x})] = \sum_{\boldsymbol{x} \in D} p(\boldsymbol{x}) f(\boldsymbol{x}). \tag{1}$$

This expectation comes up in reinforcement learning, discrete latent variable modelling (e.g. for compression), structured prediction (e.g. for translation), hard attention and many other tasks that use models with discrete operations in their computational graphs (see e.g. Jang et al. (2016)). In general, $\boldsymbol{x}$ has structure (such as a sequence), but we can treat it as a 'flat' distribution, omitting the bold notation, so $x$ has a categorical distribution over $D$ given by $p(x), x \in D$. Typically, the distribution has parameters $\boldsymbol{\theta}$, which are learnt through gradient descent. This requires estimating the gradient $\nabla_{\boldsymbol{\theta}} \mathbb{E}_{x \sim p_{\boldsymbol{\theta}}(x)}[f(x)]$, using a set of samples $S$. A gradient estimate $e(S)$ is unbiased if

$$\mathbb{E}_S[e(S)] = \nabla_{\boldsymbol{\theta}} \mathbb{E}_{x \sim p_{\boldsymbol{\theta}}(x)}[f(x)]. \tag{2}$$

The samples $S$ can be sampled independently or using alternatives such as stratified sampling which reduce variance to increase the speed of learning. In this paper, we derive an unbiased gradient estimator that reduces variance by avoiding duplicate samples, i.e. by sampling $S$ without replacement. This is challenging as samples without replacement are dependent and have marginal distributions that are different from $p(x)$. We further reduce the variance by deriving a built-in control variate, which maintains the unbiasedness and does not require additional samples.

**Related work.** Many algorithms for estimating gradients for discrete distributions have been proposed. A general and widely used estimator is REINFORCE (Williams, 1992). Biased gradients based on a continuous relaxations of the discrete distribution (known as Gumbel-Softmax or Concrete) were jointly introduced by Jang et al. (2016) and Maddison et al. (2016). These can be combined with the straight through estimator (Bengio et al., 2013) if the model requires discrete samples or be used to construct control variates for REINFORCE, as in REBAR (Tucker et al., 2017) or

---

[1]Code available at https://github.com/wouterkool/estimating-gradients-without-replacement.

RELAX (Grathwohl et al., 2018). Many other methods use control variates and other techniques to reduce the variance of REINFORCE (Paisley et al., 2012; Ranganath et al., 2014; Gregor et al., 2014; Mnih & Gregor, 2014; Gu et al., 2016; Mnih & Rezende, 2016).

Some works rely on explicit summation of the expectation, either for the marginal distribution (Titsias & Lázaro-Gredilla, 2015) or globally summing some categories while sampling from the remainder (Liang et al., 2018; Liu et al., 2019). Other approaches use a finite difference approximation to the gradient (Lorberbom et al., 2018; 2019). Yin et al. (2019) introduced ARSM, which uses multiple model evaluations where the number adapts automatically to the uncertainty.

In the structured prediction setting, there are many algorithms for optimizing a quantity under a sequence of discrete decisions, using (weak) supervision, multiple samples (or deterministic model evaluations), or a combination both (Ranzato et al., 2016; Shen et al., 2016; He et al., 2016; Norouzi et al., 2016; Bahdanau et al., 2017; Edunov et al., 2018; Leblond et al., 2018; Negrinho et al., 2018). Most of these algorithms are biased and rely on pretraining using maximum likelihood or gradually transitioning from supervised to reinforcement learning. Using Gumbel-Softmax based approaches in a sequential setting is difficult as the bias accumulates because of mixing errors (Gu et al., 2018).

## 2 PRELIMINARIES

Throughout this paper, we will denote with $B^k$ an *ordered* sample without replacement of size $k$ and with $S^k$ an *unordered* sample (of size $k$) from the categorical distribution $p$.

**Restricted distribution.** When sampling without replacement, we remove the set $C \subset D$ already sampled from the domain and we denote with $p^{D \setminus C}$ the distribution *restricted to the domain* $D \setminus C$:

$$p^{D \setminus C}(x) = \frac{p(x)}{1 - \sum_{c \in C} p(c)}, \quad x \in D \setminus C. \tag{3}$$

**Ordered sample without replacement $B^k$.** Let $B^k = (b_1, ..., b_k), b_i \in D$ be an *ordered sample without replacement*, which is generated from the distribution $p$ as follows: first, sample $b_1 \sim p$, then sample $b_2 \sim p^{D \setminus \{b_1\}}$, $b_3 \sim p^{D \setminus \{b_1, b_2\}}$, etc. i.e. elements are sampled one by one without replacement. Using this procedure, $B^k$ can be seen as a (partial) ranking according to the Plackett-Luce model (Plackett, 1975; Luce, 1959) and the probability of obtaining the vector $B^k$ is

$$p(B^k) = \prod_{i=1}^k p^{D \setminus B^{i-1}}(b_i) = \prod_{i=1}^k \frac{p(b_i)}{1 - \sum_{j < i} p(b_j)}. \tag{4}$$

We can also restrict $B^k$ to the domain $D \setminus C$, which means that $b_i \notin C$ for $i = 1, ..., k$:

$$p^{D \setminus C}(B^k) = \prod_{i=1}^k \frac{p^{D \setminus C}(b_i)}{1 - \sum_{j < i} p^{D \setminus C}(b_j)} = \prod_{i=1}^k \frac{p(b_i)}{1 - \sum_{c \in C} p(c) - \sum_{j < i} p(b_j)}. \tag{5}$$

**Unordered sample without replacement.** Let $S^k \subseteq D$ be an *unordered* sample without replacement from the distribution $p$, which can be generated simply by generating an ordered sample and discarding the order. We denote elements in the sample with $s \in S^k$ (so without index) and we write $\mathcal{B}(S^k)$ as the set of all $k!$ permutations (orderings) $B^k$ that correspond to (could have generated) $S^k$. It follows that the probability for sampling $S^k$ is given by:

$$p(S^k) = \sum_{B^k \in \mathcal{B}(S^k)} p(B^k) = \sum_{B^k \in \mathcal{B}(S^k)} \prod_{i=1}^k \frac{p(b_i)}{1 - \sum_{j < i} p(b_j)} = \left( \prod_{s \in S^k} p(s) \right) \cdot \sum_{B^k \in \mathcal{B}(S^k)} \prod_{i=1}^k \frac{1}{1 - \sum_{j < i} p(b_j)}. \tag{6}$$

The last step follows since $B^k \in \mathcal{B}(S^k)$ is an ordering of $S^k$, such that $\prod_{i=1}^k p(b_i) = \prod_{s \in S} p(s)$. Naive computation of $p(S^k)$ is $O(k!)$, but in Appendix B we show how to compute it efficiently.

When sampling from the distribution restricted to $D \setminus C$, we sample $S^k \subseteq D \setminus C$ with probability:

$$p^{D \setminus C}(S^k) = \left( \prod_{s \in S^k} p(s) \right) \cdot \sum_{B^k \in \mathcal{B}(S^k)} \prod_{i=1}^{k} \frac{1}{1 - \sum\limits_{c \in C} p(c) - \sum\limits_{j < i} p(b_j)}. \tag{7}$$

**The Gumbel-Top-$k$ trick.**   As an alternative to sequential sampling, we can also sample $B^k$ and $S^k$ by taking the top $k$ of Gumbel variables (Yellott, 1977; Vieira, 2014; Kim et al., 2016). Following notation from Kool et al. (2019c), we define the *perturbed log-probability* $g_{\phi_i} = \phi_i + g_i$, where $\phi_i = \log p(i)$ and $g_i \sim \text{Gumbel}(0)$. Then let $b_1 = \arg \max_{i \in D} g_{\phi_i}$, $b_2 = \arg \max_{i \in D \setminus \{b_1\}} g_{\phi_i}$, etc., so $B^k$ is the top $k$ of the perturbed log-probabilities *in decreasing order*. The probability of obtaining $B_k$ using this procedure is given by equation 4, so this provides an alternative sampling method which is effectively a (non-differentiable) reparameterization of sampling without replacement. For a differentiable reparameterization, see Grover et al. (2019).

It follows that taking the top $k$ perturbed log-probabilities *without order*, we obtain the unordered sample set $S^k$. This way of sampling underlies the efficient computation of $p(S^k)$ in Appendix B.

## 3   METHODOLOGY

In this section, we derive the *unordered set policy gradient estimator*: a low-variance, unbiased estimator of $\nabla_{\boldsymbol{\theta}} \mathbb{E}_{p_{\boldsymbol{\theta}}(x)}[f(x)]$ based on an unordered sample without replacement $S^k$. First, we derive the generic (non-gradient) estimator for $\mathbb{E}[f(x)]$ as the Rao-Blackwellized version of a single sample Monte Carlo estimator (and two other estimators!). Then we combine this estimator with REINFORCE (Williams, 1992) and we show how to reduce its variance using a built-in baseline.

### 3.1   RAO-BLACKWELLIZATION OF THE SINGLE SAMPLE ESTIMATOR

A very crude but simple estimator for $\mathbb{E}[f(x)]$ based on the *ordered* sample $B^k$ is to *only* use the first element $b_1$, which by definition is a sample from the distribution $p$. We define this estimator as the *single sample estimator*, which is unbiased, since

$$\mathbb{E}_{B^k \sim p(B^k)}[f(b_1)] = \mathbb{E}_{b_1 \sim p(b_1)}[f(b_1)] = \mathbb{E}_{x \sim p(x)}[f(x)]. \tag{8}$$

Discarding all but one sample, the single sample estimator is inefficient, but we can use Rao-Blackwellization (Casella & Robert, 1996) to signficantly improve it. To this end, we consider the distribution $B^k | S^k$, which is, knowing the unordered sample $S^k$, the conditional distribution over ordered samples $B^k \in \mathcal{B}(S^k)$ that could have generated $S^k$.[2] Using $B^k | S^k$, we rewrite $\mathbb{E}[f(b_1)]$ as

$$\mathbb{E}_{B^k \sim p(B^k)}[f(b_1)] = \mathbb{E}_{S^k \sim p(S^k)} \left[ \mathbb{E}_{B^k \sim p(B^k | S^k)}[f(b_1)] \right] = \mathbb{E}_{S^k \sim p(S^k)} \left[ \mathbb{E}_{b_1 \sim p(b_1 | S^k)}[f(b_1)] \right].$$

The Rao-Blackwellized version of the single sample estimator computes the inner conditional expectation exactly. Since $B^k$ is an ordering of $S^k$, we have $b_1 \in S^k$ and we can compute this as

$$\mathbb{E}_{b_1 \sim p(b_1 | S^k)}[f(b_1)] = \sum_{s \in S^k} P(b_1 = s | S^k) f(s) \tag{9}$$

where, in a slight abuse of notation, $P(b_1 = s | S^k)$ is the probability that the first sampled element $b_1$ takes the value $s$, given that the complete set of $k$ samples is $S^k$. Using Bayes' Theorem we find

$$P(b_1 = s | S^k) = \frac{p(S^k | b_1 = s) P(b_1 = s)}{p(S^k)} = \frac{p^{D \setminus \{s\}}(S^k \setminus \{s\}) p(s)}{p(S^k)}. \tag{10}$$

The step $p(S^k | b_1 = s) = p^{D \setminus \{s\}}(S^k \setminus \{s\})$ comes from analyzing sequential sampling without replacement: given that the first element sampled is $s$, the remaining elements have a distribution restricted to $D \setminus \{s\}$, so sampling $S^k$ (including $s$) given the first element $s$ is equivalent to sampling the remainder $S^k \setminus \{s\}$ from the restricted distribution, which has probability $p^{D \setminus \{s\}}(S^k \setminus \{s\})$ (see equation 7).

---

[2]Note that $B^k | S^k$ is *not* a Plackett-Luce distribution restricted to $S^k$!

**The unordered set estimator.** For notational convenience, we introduce the *leave-one-out ratio*.

**Definition 1.** The *leave-one-out ratio* of $s$ w.r.t. the set $S$ is given by $R(S^k, s) = \frac{p^{D \setminus \{s\}}(S^k \setminus \{s\})}{p(S^k)}$.

Rewriting equation 10 as $P(b_1 = s | S^k) = p(s)R(S^k, s)$ shows that the probability of sampling $s$ first, given $S^k$, is simply the unconditional probability multiplied by the leave-one-out ratio. We now define the unordered set estimator as the Rao-Blackwellized version of the single-sample estimator.

**Theorem 1.** *The* unordered set estimator, *given by*

$$e^{US}(S^k) = \sum_{s \in S^k} p(s)R(S^k, s)f(s) \tag{11}$$

*is the Rao-Blackwellized version of the (unbiased!) single sample estimator.*

*Proof.* Using $P(b_1 = s | S^k) = p(s)R(S^k, s)$ in equation 9 we have

$$\mathbb{E}_{b_1 \sim p(b_1 | S^k)}[f(b_1)] = \sum_{s \in S^k} P(b_1 = s | S^k)f(s) = \sum_{s \in S^k} p(s)R(S^k, s)f(s). \tag{12}$$

$\square$

The implication of this theorem is that the unordered set estimator, in explicit form given by equation 11, is an unbiased estimator of $\mathbb{E}[f(x)]$ since it is the Rao-Blackwellized version of the unbiased single sample estimator. Also, as expected by taking multiple samples, it has variance equal or lower than the single sample estimator by the Rao-Blackwell Theorem (Lehmann & Scheffé, 1950).

## 3.2 RAO-BLACKWELLIZATION OF OTHER ESTIMATORS

The unordered set estimator is also the result of Rao-Blackwellizing two other unbiased estimators: the *stochastic sum-and-sample* estimator and the *importance-weighted estimator*.

**The sum-and-sample estimator.** We define as *sum-and-sample estimator* any estimator that relies on the identity that for any $C \subset D$

$$\mathbb{E}_{x \sim p(x)}[f(x)] = \mathbb{E}_{x \sim p^{D \setminus C}(x)}\left[ \sum_{c \in C} p(c)f(c) + \left(1 - \sum_{c \in C} p(c)\right)f(x) \right]. \tag{13}$$

For the derivation, see Appendix C.1 or Liang et al. (2018); Liu et al. (2019). In general, a sum-and-sample estimator with a budget of $k > 1$ evaluations sums expectation terms for a set of categories $C$ (s.t. $|C| < k$) explicitly (e.g. selected by their value $f$ (Liang et al., 2018) or probability $p$ (Liu et al., 2019)), and uses $k - |C|$ (down-weighted) samples from $D \setminus C$ to estimate the remaining terms. As is noted by Liu et al. (2019), selecting $C$ such that $\frac{1 - \sum_{c \in C} p(c)}{k - |C|}$ is minimized guarantees to reduce variance compared to a standard minibatch of $k$ samples (which is equivalent to setting $C = \emptyset$). See also Fearnhead & Clifford (2003) for a discussion on selecting $C$ optimally. The ability to optimize $C$ depends on whether $p(c)$ can be computed efficiently a-priori (before sampling). This is difficult in high-dimensional settings, e.g. sequence models which compute the probability incrementally while ancestral sampling. An alternative is to select $C$ stochastically (as equation 13 holds for any $C$), and we choose $C = B^{k-1}$ to define the *stochastic sum-and-sample* estimator:

$$e^{\mathsf{SSAS}}(B^k) = \sum_{j=1}^{k-1} p(b_j)f(b_j) + \left(1 - \sum_{j=1}^{k-1} p(b_j)\right)f(b_k). \tag{14}$$

For simplicity, we consider the version that sums $k - 1$ terms here, but the following results also hold for a version that sums $k - m$ terms and uses $m$ samples (without replacement) (see Appendix C.3). Sampling without replacement, it holds that $b_k | B^{k-1} \sim p^{D \setminus B^{k-1}}$, so the unbiasedness follows from equation 13 by separating the expectation over $B^k$ into expectations over $B^{k-1}$ and $b_k | B^{k-1}$:

$$\mathbb{E}_{B^{k-1} \sim p(B^{k-1})}\left[\mathbb{E}_{b_k \sim p(b_k | B^{k-1})}\left[e^{\mathsf{SSAS}}(B^k)\right]\right] = \mathbb{E}_{B^{k-1} \sim p(B^{k-1})}[\mathbb{E}[f(x)]] = \mathbb{E}[f(x)].$$

In general, a sum-and-sample estimator reduces variance if the probability mass is concentrated on the summed categories. As typically high probability categories are sampled first, the stochastic sum-and-sample estimator sums high probability categories, similar to the estimator by Liu et al. (2019) which we refer to as the *deterministic sum-and-sample estimator*. As we show in Appendix C.2, Rao-Blackwellizing the stochastic sum-and-sample estimator also results in the unordered set estimator. This even holds for a version that uses $m$ samples and $k-m$ summed terms (see Appendix C.3), which means that the unordered set estimator has equal or lower variance than the optimal (in terms of $m$) stochastic sum-and-sample estimator, but conveniently does *not* need to choose $m$.

**The importance-weighted estimator.** The importance-weighted estimator (Vieira, 2017) is

$$e^{\text{IW}}(S^k, \kappa) = \sum_{s \in S^k} \frac{p(s)}{q(s, \kappa)} f(s). \tag{15}$$

This estimator is based on the idea of priority sampling (Duffield et al., 2007). It does *not* use the order of the sample, but assumes sampling using the Gumbel-Top-$k$ trick and requires access to $\kappa$, the $(k+1)$-th largest perturbed log-probability, which can be seen as the 'threshold' since $g_{\phi_s} > \kappa \; \forall s \in S^k$. $q(s, a) = P(g_{\phi_s} > a)$ can be interpreted as the *inclusion probability* of $s \in S^k$ (assuming a fixed threshold $a$ instead of a fixed sample size $k$). For details and a proof of unbiasedness, see Vieira (2017) or Kool et al. (2019c). As the estimator has high variance, Kool et al. (2019c) resort to *normalizing* the importance weights, resulting in biased estimates. Instead, we use Rao-Blackwellization to eliminate stochasticity by $\kappa$. Again, the result is the unordered set estimator (see Appendix D.1), which thus has equal or lower variance.

### 3.3 THE UNORDERED SET POLICY GRADIENT ESTIMATOR

Writing $p_{\boldsymbol{\theta}}$ to indicate the dependency on the model parameters $\boldsymbol{\theta}$, we can combine the unordered set estimator with REINFORCE (Williams, 1992) to obtain the *unordered set policy gradient* estimator.

**Corollary 1.** *The* unordered set policy gradient estimator, *given by*

$$e^{USPG}(S^k) = \sum_{s \in S^k} p_{\boldsymbol{\theta}}(s) R(S^k, s) \nabla_{\boldsymbol{\theta}} \log p_{\boldsymbol{\theta}}(s) f(s) = \sum_{s \in S^k} \nabla_{\boldsymbol{\theta}} p_{\boldsymbol{\theta}}(s) R(S^k, s) f(s), \tag{16}$$

*is an unbiased estimate of the policy gradient.*

*Proof.* Using REINFORCE (Williams, 1992) combined with the unordered set estimator we find:

$$\nabla_{\boldsymbol{\theta}} \mathbb{E}_{p_{\boldsymbol{\theta}}(x)}[f(x)] = \mathbb{E}_{p_{\boldsymbol{\theta}}(x)}[\nabla_{\boldsymbol{\theta}} \log p_{\boldsymbol{\theta}}(x) f(x)] = \mathbb{E}_{S^k \sim p_{\boldsymbol{\theta}}(S^k)} \left[ \sum_{s \in S^k} p_{\boldsymbol{\theta}}(s) R(S^k, s) \nabla_{\boldsymbol{\theta}} \log p_{\boldsymbol{\theta}}(s) f(s) \right].$$

$\square$

**Variance reduction using a built-in control variate.** The variance of REINFORCE can be reduced by subtracting a baseline from $f$. When taking multiple samples (with replacement), a simple and effective baseline is to take the mean of other (independent!) samples (Mnih & Rezende, 2016). Sampling without replacement, we can use the same idea to construct a baseline based on the other samples, but we have to correct for the fact that the samples are *not* independent.

**Theorem 2.** *The* unordered set policy gradient estimator with baseline, *given by*

$$e^{USPGBL}(S^k) = \sum_{s \in S^k} \nabla_{\boldsymbol{\theta}} p_{\boldsymbol{\theta}}(s) R(S^k, s) \left( f(s) - \sum_{s' \in S^k} p_{\boldsymbol{\theta}}(s') R^{D \backslash \{s\}}(S^k, s') f(s') \right), \tag{17}$$

*where*

$$R^{D \backslash \{s\}}(S^k, s') = \frac{p_{\boldsymbol{\theta}}^{D \backslash \{s, s'\}}(S^k \backslash \{s, s'\})}{p_{\boldsymbol{\theta}}^{D \backslash \{s\}}(S^k \backslash \{s\})} \tag{18}$$

*is the* second order leave-one-out ratio, *is an unbiased estimate of the policy gradient.*

*Proof.* See Appendix E.1. □

This theorem shows how to include a built-in baseline based on *dependent* samples (without replacement), without introducing bias. By having a built-in baseline, the value $f(s)$ for sample $s$ is compared against an estimate of its expectation $\mathbb{E}[f(s)]$, based on the other samples. The difference is an estimate of the *advantage* (Sutton & Barto, 2018), which is positive if the sample $s$ is 'better' than average, causing $p_{\boldsymbol{\theta}}(s)$ to be increased (reinforced) through the sign of the gradient, and vice versa. By sampling without replacement, the unordered set estimator forces the estimator to compare different alternatives, and reinforces the best among them.

**Including the pathwise derivative.** So far, we have only considered the scenario where $f$ does not depend on $\boldsymbol{\theta}$. If $f$ does depend on $\boldsymbol{\theta}$, for example in a VAE (Kingma & Welling, 2014; Rezende et al., 2014), then we use the notation $f_{\boldsymbol{\theta}}$ and we can write the gradient (Schulman et al., 2015) as

$$\nabla_{\boldsymbol{\theta}}\mathbb{E}_{p_{\boldsymbol{\theta}}(x)}[f_{\boldsymbol{\theta}}(x)] = \mathbb{E}_{p_{\boldsymbol{\theta}}(x)}[\nabla_{\boldsymbol{\theta}}\log p_{\boldsymbol{\theta}}(x)f_{\boldsymbol{\theta}}(x) + \nabla_{\boldsymbol{\theta}}f_{\boldsymbol{\theta}}(x)]. \tag{19}$$

The additional second ('pathwise') term can be estimated (using the same samples) with the standard unordered set estimator. This results in the *full* unordered set policy gradient estimator:

$$
\begin{aligned}
e^{\text{FUSPG}}(S^k) &= \sum_{s \in S^k} \nabla_{\boldsymbol{\theta}}p_{\boldsymbol{\theta}}(s)R(S^k, s)f_{\boldsymbol{\theta}}(s) + \sum_{s \in S^k} p_{\boldsymbol{\theta}}(s)R(S^k, s)\nabla_{\boldsymbol{\theta}}f_{\boldsymbol{\theta}}(s) \\
&= \sum_{s \in S^k} R(S^k, s)\nabla_{\boldsymbol{\theta}}\left(p_{\boldsymbol{\theta}}(s)f_{\boldsymbol{\theta}}(s)\right)
\end{aligned}
\tag{20}
$$

Equation 20 is straightforward to implement using an automatic differentiation library. We can also include the baseline (as in equation 17) but we must make sure to call STOP_GRADIENT (DETACH in PyTorch) on the baseline (but not on $f_{\boldsymbol{\theta}}(s)$!). Importantly, we should *never* track gradients through the leave-one-out ratio $R(S^k, s)$ which means it can be efficiently computed in pure inference mode.

**Scope & limitations.** We can use the unordered set estimator for any discrete distribution from which we can sample without replacement, by treating it as a univariate categorical distribution over its domain. This includes sequence models, from which we can sample using Stochastic Beam Search (Kool et al., 2019c), as well as multivariate categorical distributions which can also be treated as sequence models (see Section 4.2). In the presence of continuous variables or a stochastic function $f$, we may separate this stochasticity from the stochasticity over the discrete distribution, as in Lorberbom et al. (2019). The computation of the leave-one-out ratios adds some overhead, although they can be computed efficiently, even for large $k$ (see Appendix B). For a moderately sized model, the costs of model evaluation and backpropagation dominate the cost of computing the estimator.

### 3.4 RELATION TO OTHER MULTI-SAMPLE ESTIMATORS

**Relation to Murthy's estimator.** We found out that the 'vanilla' unordered set estimator (equation 11) is actually a special case of the estimator by Murthy (1957), known in statistics literature for estimation of a population total $\Theta = \sum_{i \in D} y_i$. Using $y_i = p(i)f(i)$, we have $\Theta = \mathbb{E}[f(i)]$, so Murthy's estimator can be used to estimate expectations (see equation 11). Murthy derives the estimator by 'unordering' a convex combination of Raj (1956) estimators, which, using $y_i = p(i)f(i)$, are stochastic sum-and-sample estimators in our analogy.

Murthy (1957) also provides an unbiased estimator of the variance, which may be interesting for future applications. Since Murthy's estimator can be used with *arbitrary* sampling distribution, it is straightforward to derive importance-sampling versions of our estimators. In particular, we can sample $S$ without replacement using $q(x) > 0, x \in D$, and use equations 11, 16, 17 and 20, as long as we compute the leave-one-out ratio $R(S^k, s)$ using $q$.

While part of our derivation coincides with Murthy (1957), we are not aware of previous work using this estimator to estimate expectations. Additionally, we discuss practical computation of $p(S)$ (Appendix B), we show the relation to the importance-weighted estimator, and we provide the extension to estimating policy gradients, especially including a built-in baseline without adding bias.

**Relation to the empirical risk estimator.** The empirical risk loss (Edunov et al., 2018) estimates the expectation in equation 1 by summing only a subset $S$ of the domain, using *normalized* probabilities $\hat{p}_{\boldsymbol{\theta}}(s) = \frac{p_{\boldsymbol{\theta}}(s)}{\sum_{s' \in S} p_{\boldsymbol{\theta}}(s)}$. Using this loss, the (biased) estimate of the gradient is given by

$$e^{\text{RISK}}(S^k) = \sum_{s \in S^k} \nabla_{\boldsymbol{\theta}} \left( \frac{p_{\boldsymbol{\theta}}(s)}{\sum_{s' \in S^k} p_{\boldsymbol{\theta}}(s')} \right) f(s). \tag{21}$$

The risk estimator is similar to the unordered set policy gradient estimator, with two important differences: 1) the individual terms are normalized by the total probability mass rather than the leave-one-out ratio and 2) the gradient w.r.t. the normalization factor is taken into account. As a result, samples 'compete' for probability mass and only the best can be reinforced. This has the same effect as using a built-in baseline, which we prove in the following theorem.

**Theorem 3.** *By taking the gradient w.r.t. the normalization factor into account, the risk estimator has a built-in baseline, which means it can be written as*

$$e^{RISK}(S^k) = \sum_{s \in S^k} \nabla_{\boldsymbol{\theta}} p_{\boldsymbol{\theta}}(s) \frac{1}{\sum_{s'' \in S^k} p_{\boldsymbol{\theta}}(s'')} \left( f(s) - \sum_{s' \in S^k} p_{\boldsymbol{\theta}}(s') \frac{1}{\sum_{s'' \in S^k} p_{\boldsymbol{\theta}}(s'')} f(s') \right). \tag{22}$$

*Proof.* See Appendix F.1 □

This theorem highlights the similarity between the biased risk estimator and our unbiased estimator (equation 17), and suggests that their only difference is the weighting of terms. Unfortunately, the implementation by Edunov et al. (2018) has more sources of bias (e.g. length normalization), which are not compatible with our estimator. However, we believe that our analysis helps analyze the bias of the risk estimator and is a step towards developing unbiased estimators for structured prediction.

**Relation to VIMCO.** VIMCO (Mnih & Rezende, 2016) is an estimator that uses $k$ samples (with replacement) to optimize an objective of the form $\log \frac{1}{k} \sum_i f(x_i)$, which is a multi-sample stochastic lower bound in the context of variational inference. VIMCO reduces the variance by using a *local* baseline for each of the $k$ samples, based on the other $k-1$ samples. While we do not have a log term, as our goal is to optimize general $\mathbb{E}[f(x)]$, we adopt the idea of forming a baseline based on the other samples, and we define *REINFORCE with replacement* (with built-in baseline) as the estimator that computes the gradient estimate using samples with replacement $X^k = (x_1, ..., x_k)$ as

$$e^{\text{RFWR}}(X^k) = \frac{1}{k} \sum_{i=1}^{k} \nabla_{\boldsymbol{\theta}} \log p_{\boldsymbol{\theta}}(x_i) \left( f(x_i) - \frac{1}{k-1} \sum_{j \neq i} f(x_j) \right). \tag{23}$$

This estimator is unbiased, as $\mathbb{E}_{x_i, x_j}[\nabla_{\boldsymbol{\theta}} \log p_{\boldsymbol{\theta}}(x_i) f(x_j)] = 0$ for $i \neq j$ (see also Kool et al. (2019b)). We think of the unordered set estimator as the without-replacement version of this estimator, which weights terms by $p_{\boldsymbol{\theta}}(s)R(S^k, s)$ instead of $\frac{1}{k}$. This puts more weight on higher probability elements to compensate for sampling without replacement. If probabilities are small and (close to) uniform, there are (almost) no duplicate samples and the weights will be close to $\frac{1}{k}$, so the gradient estimate of the with- and without-replacement versions are similar.

**Relation to ARSM.** ARSM (Yin et al., 2019) also uses multiple evaluations ('pseudo-samples') of $p_{\boldsymbol{\theta}}$ and $f$. This can be seen as similar to sampling without replacement, and the estimator also has a built-in control variate. Compared to ARSM, our estimator allows direct control over the computational cost (through the sample size $k$) and has wider applicability, for example it also applies to multivariate categorical variables with different numbers of categories per dimension.

**Relation to stratified/systematic sampling.** Our estimator aims to reduce variance by changing the sampling distribution for multiple samples by sampling without replacement. There are alternatives, such as using stratified or systematic sampling (see, e.g. Douc & Cappé (2005)). Both partition the domain $D$ into $k$ strata and take a single sample from each stratum, where systematic sampling uses common random numbers for each stratum. In applications involving high-dimensional or structured domains, it is unclear how to partition the domain and how to sample from each partition. Additionally, as samples are *not* independent, it is non-trivial to include a built-in baseline, which we find is a key component that makes our estimator perform well.

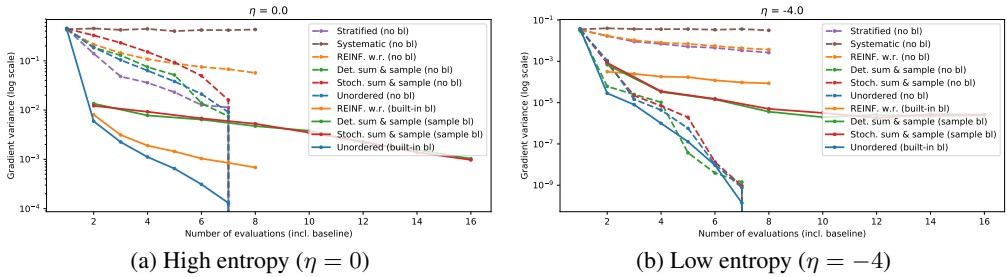

(a) High entropy ($\eta = 0$)  (b) Low entropy ($\eta = -4$)

Figure 1: Bernoulli gradient variance (on log scale) as a function of the number of model evaluations (including baseline evaluations, so the sum-and-sample estimators with sampled baselines use twice as many evaluations). Note that for some estimators, the variance is 0 (log variance $-\infty$) for $k = 8$.

## 4 EXPERIMENTS

### 4.1 BERNOULLI TOY EXPERIMENT

We use the code by Liu et al. (2019) to reproduce their Bernoulli toy experiment. Given a vector $\mathbf{p} = (0.6, 0.51, 0.48)$ the goal is to minimize the loss $\mathcal{L}(\eta) = \mathbb{E}_{x_1,x_2,x_3 \sim \mathrm{Bern}(\sigma(\eta))} \left[ \sum_{i=1}^{3} (x_i - p_i)^2 \right]$. Here $x_1, x_2, x_3$ are i.i.d. from the Bernoulli($\sigma(\eta)$) distribution, parameterized by a scalar $\eta \in \mathbb{R}$, where $\sigma(\eta) = (1 + \exp(-\eta))^{-1}$ is the sigmoid function. We compare different estimators, with and without baseline (either 'built-in' or using additional samples, referred to as REINFORCE+ in Liu et al. (2019)). We report the (log-)variance of the scalar gradient $\frac{\partial \mathcal{L}}{\partial \eta}$ as a function of the number of model evaluations, which is twice as high when using a sampled baseline (for each term).

As can be seen in Figure 1, the unordered set estimator is the only estimator that has consistently the lowest (or comparable) variance in both the high ($\eta = 0$) and low entropy ($\eta = -4$) regimes and for different number of samples/model evaluations. This suggests that it combines the advantages of the other estimators. We also ran the actual optimization experiment, where with as few as $k = 3$ samples the trajectory was indistinguishable from using the exact gradient (see Liu et al. (2019)).

### 4.2 CATEGORICAL VARIATIONAL AUTO-ENCODER

We use the code from Yin et al. (2019) to train a *categorical* Variational Auto-Encoder (VAE) with 20 dimensional latent space, with 10 categories per dimension (details in Appendix G.1). To use our estimator, we treat this as a single factorized distribution with $10^{20}$ categories from which we can sample without replacement using Stochastic Beam Search (Kool et al., 2019c), sequentially sampling each dimension as if it were a sequence model. We also perform experiments with $10^2$ latent space, which provides a lower entropy setting, to highlight the advantage of our estimator.

**Measuring the variance.** In Table 1, we report the variance of different gradient estimators with $k = 4$ samples, evaluated on a trained model. The unordered set estimator has the lowest variance in both the small and large domain (low and high entropy) setting, being on-par with the best of the (stochastic[3]) sum-and-sample estimator and REINFORCE with replacement[4]. This confirms the toy experiment, suggesting that the unordered set estimator provides the best of both estimators. In Appendix G.2 we repeat the same experiment at different stages of training, with similar results.

---

[3]We cannot use the deterministic version by Liu et al. (2019) since we cannot select the top $k$ categories.
[4]We cannot compare against VIMCO (Mnih & Rezende, 2016) as it optimizes a different objective.

Table 1: VAE gradient log-variance of different unbiased estimators with $k = 4$ samples.

| Domain | ARSM | RELAX | REINFORCE (no bl) | (sample bl) | Sum & sample (no bl) | (sample bl) | REINF. w.r. (built-in bl) | Unordered (built-in bl) |
|---|---|---|---|---|---|---|---|---|
| Small $10^2$ | 13.45 | 11.67 | 11.52 | 7.49 | **6.29** | **6.29** | 6.65 | **6.29** |
| Large $10^{20}$ | 15.55 | 15.86 | 13.81 | 8.48 | 13.77 | 8.44 | **7.06** | **7.05** |

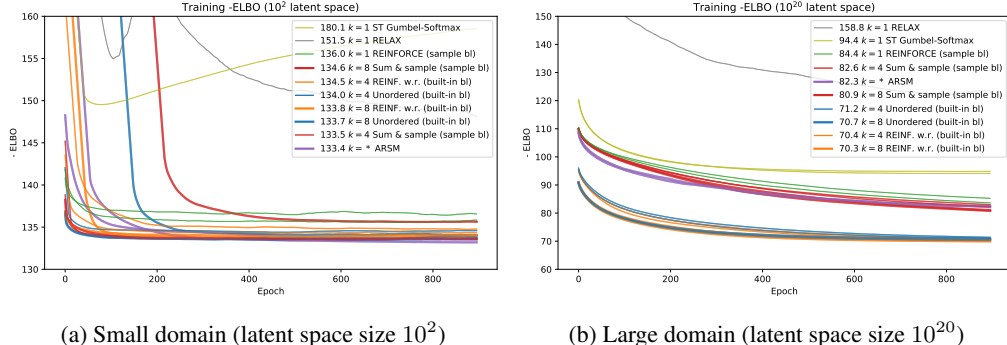

(a) Small domain (latent space size $10^2$)  (b) Large domain (latent space size $10^{20}$)

Figure 2: VAE smoothed training curves (-ELBO) of two independent runs when training with different estimators with $k = 1$, 4 or 8 (thicker lines) samples (ARSM has a variable number). Some lines coincide, so we sort the legend by the lowest -ELBO achieved and report this value.

**ELBO optimization.** We use different estimators to optimize the ELBO (details in Appendix G.1). Additionally to the baselines by Yin et al. (2019) we compare against REINFORCE with replacement and the stochastic sum-and-sample estimator. In Figure 2 we observe that our estimator performs on par with REINFORCE with replacement (and built-in baseline, equation 23) and outperforms other estimators in at least one of the settings. There are a lot of other factors, e.g. exploration that may explain why we do not get a strictly better result despite the lower variance. We note some overfitting (see validation curves in Appendix G.2), but since our goal is to show improved optimization, and to keep results directly comparable to Yin et al. (2019), we consider regularization a separate issue outside the scope of this paper. These results are using MNIST binarized by a threshold of 0.5. In Appendix G.2 we report results using the standard binarized MNIST dataset from Salakhutdinov & Murray (2008).

### 4.3 STRUCTURED PREDICTION FOR THE TRAVELLING SALESMAN PROBLEM

To show the wide applicability of our estimator, we consider the structured prediction task of predicting routes (sequences) for the Travelling Salesman Problem (TSP) (Vinyals et al., 2015; Bello et al., 2016; Kool et al., 2019a). We use the code by Kool et al. (2019a)[5] to reproduce their TSP experiment with 20 nodes. For details, see Appendix H.

We implement REINFORCE with replacement (and built-in baseline) as well as the stochastic sum-and-sample estimator and our estimator, using Stochastic Beam Search (Kool et al., 2019c) for sampling. Also, we include results using the biased normalized importance-weighted policy gradient estimator with built-in baseline (derived in Kool et al. (2019b), see Appendix D.2). Additionally, we compare against REINFORCE with greedy rollout baseline (Rennie et al., 2017) used by Kool et al. (2019c) and a batch-average baseline. For reference, we also include the biased risk estimator, either 'sampling' using stochastic or deterministic beam search (as in Edunov et al. (2018)).

In Figure 3a, we compare training progress (measured on the validation set) as a function of the number of training steps, where we divide the batch size by $k$ to keep the total number of samples equal. Our estimator outperforms REINFORCE with replacement, the stochastic sum-and-sample estimator and the strong greedy rollout baseline (which uses additional baseline model evaluations) and performs on-par with the biased risk estimator. In Figure 3b, we plot the same results against the number of instances, which shows that, compared to the single sample estimators, we can train with less data and less computational cost (as we only need to run the encoder once for each instance).

---

[5]https://github.com/wouterkool/attention-learn-to-route

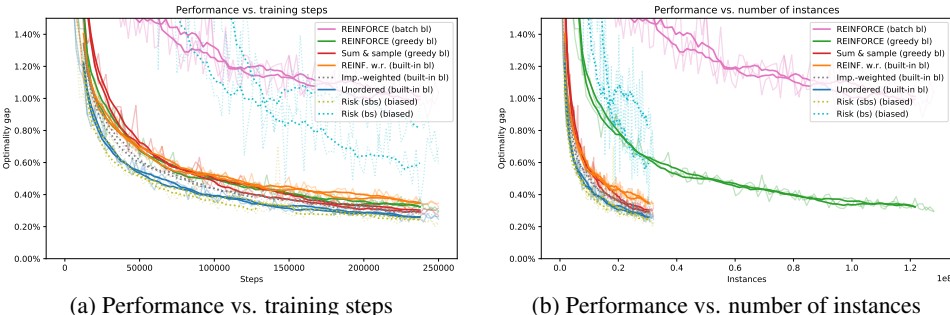

(a) Performance vs. training steps      (b) Performance vs. number of instances

Figure 3: TSP validation set optimality gap measured during training. Raw results are light, smoothed results are darker (2 random seeds). We compare our estimator against different unbiased and biased (dotted) multi-sample estimators and against single-sample REINFORCE, with batch-average or greedy rollout baseline.

## 5 DISCUSSION

We introduced the unordered set estimator, a low-variance, unbiased gradient estimator based on sampling without replacement, which can be used as an alternative to the popular biased Gumbel-Softmax estimator (Jang et al., 2016; Maddison et al., 2016). Our estimator is the result of Rao-Blackwellizing three existing estimators, which guarantees equal or lower variance, and is closely related to a number of other estimators. It has wide applicability, is parameter free (except for the sample size $k$) and has competitive performance to the best of alternatives in both high and low entropy regimes.

In our experiments, we found that REINFORCE *with* replacement, with multiple samples and a built-in baseline as inspired by VIMCO (Mnih & Rezende, 2016), is a simple yet strong estimator which has performance similar to our estimator in the high entropy setting. We are not aware of any recent work on gradient estimators for discrete distributions that has considered this estimator as baseline, while it may be often preferred given its simplicity. In future work, we want to investigate if we can apply our estimator to estimate gradients 'locally' (Titsias & Lázaro-Gredilla, 2015), as locally we have a smaller domain and expect more duplicate samples.

### ACKNOWLEDGMENTS

This research was funded by ORTEC. We would like to thank anonymous reviewers for their feedback that helped improve the paper.

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

## A  NOTATION

Throughout this appendix we will use the following notation from Maddison et al. (2014):

$$e_\phi(g) = \exp(-g + \phi)$$
$$F_\phi(g) = \exp(-\exp(-g + \phi))$$
$$f_\phi(g) = e_\phi(g)F_\phi(g).$$

This means that $F_\phi(g)$ is the CDF and $f_\phi(g)$ the PDF of the Gumbel($\phi$) distribution. Additionally we will use the identities by Maddison et al. (2014):

$$F_\phi(g)F_\gamma(g) = F_{\log(\exp(\phi)+\exp(\gamma))}(g) \tag{24}$$

$$\int_{g=a}^b e_\gamma(g)F_\phi(g)\partial g = (F_\phi(b) - F_\phi(a))\frac{\exp(\gamma)}{\exp(\phi)}. \tag{25}$$

Also, we will use the following notation, definitions and identities (see Kool et al. (2019c)):

$$\phi_i = \log p(i) \tag{26}$$

$$\phi_S = \log \sum_{i \in S} p(i) = \log \sum_{i \in S} \exp \phi_i \tag{27}$$

$$\phi_{D\setminus S} = \log \sum_{i \in D\setminus S} p(i) = \log\left(1 - \sum_{i \in S} p(i)\right) = \log(1 - \exp(\phi_S)) \tag{28}$$

$$G_{\phi_i} \sim \text{Gumbel}(\phi_i) \tag{29}$$

$$G_{\phi_S} = \max_{i \in S} G_{\phi_i} \sim \text{Gumbel}(\phi_S) \tag{30}$$

For a proof of equation 30, see Maddison et al. (2014).

## B  COMPUTATION OF $p(S^k)$, $p^{D\setminus C}(S \setminus C)$ AND $R(S^k, s)$

We can sample the set $S^k$ from the Plackett-Luce distribution using the Gumbel-Top-$k$ trick by drawing Gumbel variables $G_{\phi_i} \sim \text{Gumbel}(\phi_i)$ for each element and returning the indices of the $k$ largest Gumbels. If we ignore the ordering, this means we will obtain the set $S^k$ if $\min_{i \in S^k} G_{\phi_i} > \max_{i \in D\setminus S^k} G_{\phi_i}$. Omitting the superscript $k$ for clarity, we can use the Gumbel-Max trick, i.e. that $G_{\phi_{D\setminus S}} = \max_{i \notin S} G_{\phi_i} \sim \text{Gumbel}(\phi_{D\setminus S})$ (equation 30) and marginalize over $G_{\phi_{D\setminus S}}$:

$$p(S) = P(\min_{i \in S} G_{\phi_i} > G_{\phi_{D\setminus S}})$$
$$= P(G_{\phi_i} > G_{\phi_{D\setminus S}}, i \in S)$$
$$= \int_{g_{\phi_{D\setminus S}}=-\infty}^{\infty} f_{\phi_{D\setminus S}}(g_{\phi_{D\setminus S}})P(G_{\phi_i} > g_{\phi_{D\setminus S}}, i \in S)\partial g_{\phi_{D\setminus S}}$$
$$= \int_{g_{\phi_{D\setminus S}}=-\infty}^{\infty} f_{\phi_{D\setminus S}}(g_{\phi_{D\setminus S}})\prod_{i \in S}\left(1 - F_{\phi_i}(g_{\phi_{D\setminus S}})\right)\partial g_{\phi_{D\setminus S}} \tag{31}$$
$$= \int_{u=0}^1 \prod_{i \in S}\left(1 - F_{\phi_i}\left(F_{\phi_{D\setminus S}}^{-1}(u)\right)\right)\partial u \tag{32}$$

Here we have used a change of variables $u = F_{\phi_{D\setminus S}}(g_{\phi_{D\setminus S}})$. This expression can be efficiently numerically integrated (although another change of variables may be required for numerical stability depending on the values of $\phi$).

**Exact computation in $O(2^k)$.** The integral in equation 31 can be computed exactly using the identity

$$\prod_{i \in S}(a_i - b_i) = \sum_{C \subseteq S}(-1)^{|C|}\prod_{i \in C} b_i \prod_{i \in S\setminus C} a_i$$

which gives

$$
\begin{aligned}
p(S) &= \int_{g_{\phi_{D\setminus S}}=-\infty}^{\infty} f_{\phi_{D\setminus S}}(g_{\phi_{D\setminus S}}) \prod_{i\in S} \left(1 - F_{\phi_i}(g_{\phi_{D\setminus S}})\right) \partial g_{\phi_{D\setminus S}} \\
&= \sum_{C\subseteq S} (-1)^{|C|} \int_{g_{\phi_{D\setminus S}}=-\infty}^{\infty} f_{\phi_{D\setminus S}}(g_{\phi_{D\setminus S}}) \prod_{i\in C} F_{\phi_i}(g_{\phi_{D\setminus S}}) \prod_{i\in S\setminus C} 1 \partial g_{\phi_{D\setminus S}} \\
&= \sum_{C\subseteq S} (-1)^{|C|} \int_{g_{\phi_{D\setminus S}}=-\infty}^{\infty} e_{\phi_{D\setminus S}}(g_{\phi_{D\setminus S}}) F_{\phi_{D\setminus S}}(g_{\phi_{D\setminus S}}) F_{\phi_C}(g_{\phi_{D\setminus S}}) \partial g_{\phi_{D\setminus S}} \\
&= \sum_{C\subseteq S} (-1)^{|C|} \int_{g_{\phi_{D\setminus S}}=-\infty}^{\infty} e_{\phi_{D\setminus S}}(g_{\phi_{D\setminus S}}) F_{\phi_{(D\setminus S)\cup C}}(g_{\phi_{D\setminus S}}) \partial g_{\phi_{D\setminus S}} \\
&= \sum_{C\subseteq S} (-1)^{|C|}(1-0) \frac{\exp(\phi_{D\setminus S})}{\exp(\phi_{(D\setminus S)\cup C})} \\
&= \sum_{C\subseteq S} (-1)^{|C|} \frac{1 - \sum_{i\in S} p(i)}{1 - \sum_{i\in S\setminus C} p(i)}.
\end{aligned}
\tag{33}
$$

**Computation of $p^{D\setminus C}(S\setminus C)$.** When using the Gumbel-Top-$k$ trick over the restricted domain $D\setminus C$, we do *not* need to renormalize the log-probabilities $\phi_s, s\in D\setminus C$ since the Gumbel-Top-$k$ trick applies to unnormalized log-probabilities. Also, assuming $C\subseteq S^k$, it holds that $(D\setminus C)\setminus(S\setminus C) = D\setminus S$. This means that we can compute $p^{D\setminus C}(S\setminus C)$ similar to equation 31:

$$
\begin{aligned}
p^{D\setminus C}(S\setminus C) &= P(\min_{i\in S\setminus C} G_{\phi_i} > G_{\phi_{(D\setminus C)\setminus(S\setminus C)}}) \\
&= P(\min_{i\in S\setminus C} G_{\phi_i} > G_{\phi_{D\setminus S}}) \\
&= \int_{g_{\phi_{D\setminus S}}=-\infty}^{\infty} f_{\phi_{D\setminus S}}(g_{\phi_{D\setminus S}}) \prod_{i\in S\setminus C} \left(1 - F_{\phi_i}(g_{\phi_{D\setminus S}})\right) \partial g_{\phi_{D\setminus S}}.
\end{aligned}
\tag{34}
$$

**Computation of $R(S^k, s)$.** Note that, using equation 10, it holds that

$$
\sum_{s\in S^k} \frac{p^{D\setminus\{s\}}(S^k\setminus\{s\})p(s)}{p(S^k)} = \sum_{s\in S^k} P(b_1 = s|S^k) = 1
$$

from which it follows that

$$
p(S^k) = \sum_{s\in S^k} p^{D\setminus\{s\}}(S^k\setminus\{s\})p(s)
$$

such that

$$
R(S^k, s) = \frac{p^{D\setminus\{s\}}(S^k\setminus\{s\})}{p(S^k)} = \frac{p^{D\setminus\{s\}}(S^k\setminus\{s\})}{\sum_{s'\in S^k} p^{D\setminus\{s'\}}(S^k\setminus\{s'\})p(s')}.
\tag{35}
$$

This means that, to compute the leave-one-out ratio for all $s\in S^k$, we only need to compute $p^{D\setminus\{s\}}(S^k\setminus\{s\})$ for $s\in S^k$. When using the numerical integration or summation in $O(2^k)$, we can reuse computation, whereas using the naive method, the cost is $O(k\cdot(k-1)!) = O(k!)$, making the total computational cost comparable to computing just $p(S^k)$, and the same holds when computing the 'second-order' leave one out ratios for the built-in baseline (equation 17).

**Details of numerical integration.** For computation of the leave-one-out ratio (equation 35) for large $k$ we can use the numerical integration, where we need to compute equation 34 with $C = \{s\}$. For this purpose, we rewrite the integral as

$$p^{D\setminus C}(S \setminus C) = \int_{g_{\phi_{D\setminus S}}=-\infty}^{\infty} f_{\phi_{D\setminus S}}(g_{\phi_{D\setminus S}}) \prod_{i\in S\setminus C} \left(1 - F_{\phi_i}(g_{\phi_{D\setminus S}})\right) \partial g_{\phi_{D\setminus S}}$$

$$= \int_{u=0}^{1} \prod_{i\in S\setminus C} \left(1 - F_{\phi_i}\left(F_{\phi_{D\setminus S}}^{-1}(u)\right)\right) \partial u$$

$$= \int_{u=0}^{1} \prod_{i\in S\setminus C} \left(1 - u^{\exp(\phi_i - \phi_{D\setminus S})}\right) \partial u$$

$$= \exp(b) \cdot \int_{v=0}^{1} v^{\exp(b)-1} \prod_{i\in S\setminus C} \left(1 - v^{\exp(\phi_i - \phi_{D\setminus S}+b)}\right) \partial v$$

$$= \exp(a + \phi_{D\setminus S}) \cdot \int_{v=0}^{1} v^{\exp(a+\phi_{D\setminus S})-1} \prod_{i\in S\setminus C} \left(1 - v^{\exp(\phi_i + a)}\right) \partial v.$$

Here we have used change of variables $v = u^{exp(-b)}$ and $a = b - \phi_{D\setminus S}$. This form allows to compute the integrands efficiently, as

$$\prod_{i\in S\setminus C} \left(1 - v^{\exp(\phi_i+a)}\right) = \frac{\prod_{i\in S}\left(1 - v^{\exp(\phi_i+a)}\right)}{\prod_{i\in C}\left(1 - v^{\exp(\phi_i+a)}\right)}$$

where the numerator only needs to computed once, and, since $C = \{s\}$ when computing equation 35, the denominator only consists of a single term.

The choice of $a$ may depend on the setting, but we found that $a = 5$ is a good default option which leads to an integral that is generally smooth and can be accurately approximated using the trapezoid rule. We compute the integrands in logarithmic space and sum the terms using the stable LOGSUMEXP trick. In our code we provide an implementation which also computes all second-order leave-one-out ratios efficiently.

## C  THE SUM-AND-SAMPLE ESTIMATOR

### C.1  UNBIASEDNESS OF THE SUM-AND-SAMPLE ESTIMATOR

We show that the sum-and-sample estimator is unbiased for any set $C \subset D$ (see also Liang et al. (2018); Liu et al. (2019)):

$$\mathbb{E}_{x\sim p^{D\setminus C}(x)}\left[\sum_{c\in C} p(c)f(c) + \left(1 - \sum_{x\in C} p(c)\right)f(x)\right]$$

$$= \sum_{c\in C} p(c)f(c) + \left(1 - \sum_{c\in C} p(c)\right)\mathbb{E}_{x\sim p^{D\setminus C}(x)}[f(x)]$$

$$= \sum_{c\in C} p(c)f(c) + \left(1 - \sum_{c\in C} p(c)\right)\sum_{x\in D\setminus C}\frac{p(x)}{1 - \sum_{c\in C} p(c)}f(x)$$

$$= \sum_{c\in C} p(c)f(c) + \sum_{x\in D\setminus C} p(x)f(x)$$

$$= \sum_{x\in D} p(x)f(x)$$

$$= \mathbb{E}_{x\sim p(x)}[f(x)]$$

## C.2 RAO-BLACKWELLIZATION OF THE STOCHASTIC SUM-AND-SAMPLE ESTIMATOR

In this section we give the proof that Rao-Blackwellizing the stochastic sum-and-sample estimator results in the unordered set estimator.

**Theorem 4.** *Rao-Blackwellizing the stochastic sum-and-sample estimator results in the unordered set estimator, i.e.*

$$\mathbb{E}_{B^k \sim p(B^k | S^k)} \left[ \sum_{j=1}^{k-1} p(b_j) f(b_j) + \left( 1 - \sum_{j=1}^{k-1} p(b_j) \right) f(b_k) \right] = \sum_{s \in S^k} p(s) R(S^k, s) f(s). \quad (36)$$

*Proof.* To give the proof, we first prove three Lemmas.

**Lemma 1.**

$$P(b_k = s | S^k) = \frac{p(S^k \setminus \{s\})}{p(S^k)} \frac{p(s)}{1 - \sum_{s' \in S^k \setminus \{s\}} p(s')} \quad (37)$$

*Proof.* Similar to the derivation of $P(b_1 = s | S^k)$ (equation 10 in the main paper), we can write:

$$
\begin{aligned}
P(b_k = s | S^k) &= \frac{P(S^k \cap b_k = s)}{p(S^k)} \\
&= \frac{p(S^k \setminus \{s\}) p^{D \setminus (S^k \setminus \{s\})}(s)}{p(S^k)} \\
&= \frac{p(S^k \setminus \{s\})}{p(S^k)} \frac{p(s)}{1 - \sum_{s' \in S^k \setminus \{s\}} p(s')}.
\end{aligned}
$$

The step from the first to the second row comes from analyzing the event $S^k \cap b_k = s$ using sequential sampling: to sample $S^k$ (including $s$) with $s$ being the $k$-th element means that we should first sample $S^k \setminus \{s\}$ (in any order), and then sample $s$ from the distribution restricted to $D \setminus (S^k \setminus \{s\})$. $\quad \square$

**Lemma 2.**

$$p(S) + p(S \setminus \{s\}) \frac{1 - \sum_{s' \in S} p(s')}{1 - \sum_{s' \in S \setminus \{s\}} p(s')} = p^{D \setminus \{s\}}(S \setminus \{s\}) \quad (38)$$

Dividing equation 33 by $1 - \sum_{s' \in S} p(s')$ on both sides, we obtain

*Proof.*

$$\frac{p(S)}{1 - \sum_{s' \in S} p(s')}$$

$$= \sum_{C \subseteq S} (-1)^{|C|} \frac{1}{1 - \sum_{s' \in S \setminus C} p(s')}$$

$$= \sum_{C \subseteq S \setminus \{s\}} \left( (-1)^{|C|} \frac{1}{1 - \sum_{s' \in S \setminus C} p(s')} + (-1)^{|C \cup \{s\}|} \frac{1}{1 - \sum_{s' \in S \setminus (C \cup \{s\})} p(s')} \right)$$

$$= \sum_{C \subseteq S \setminus \{s\}} (-1)^{|C|} \frac{1}{1 - \sum_{s' \in S \setminus C} p(s')} + \sum_{C \subseteq S \setminus \{s\}} (-1)^{|C \cup \{s\}|} \frac{1}{1 - \sum_{s' \in S \setminus (C \cup \{s\})} p(s')}$$

$$= \sum_{C \subseteq S \setminus \{s\}} (-1)^{|C|} \frac{1}{1 - p(s) - \sum_{s' \in (S \setminus \{s\}) \setminus C} p(s')} - \sum_{C \subseteq S \setminus \{s\}} (-1)^{|C|} \frac{1}{1 - \sum_{s' \in (S \setminus \{s\}) \setminus C} p(s')}$$

$$= \frac{1}{1 - p(s)} \sum_{C \subseteq S \setminus \{s\}} (-1)^{|C|} \frac{1}{1 - \sum_{s' \in (S \setminus \{s\}) \setminus C} \frac{p(s')}{1-p(s)}} - \frac{p(S \setminus \{s\})}{1 - \sum_{s' \in S \setminus \{s\}} p(s')}$$

$$= \frac{1}{1 - p(s)} \frac{p^{D \setminus \{s\}}(S \setminus \{s\})}{1 - \sum_{s' \in S \setminus \{s\}} \frac{p(s')}{1-p(s)}} - \frac{p(S \setminus \{s\})}{1 - \sum_{s' \in S \setminus \{s\}} p(s')}$$

$$= \frac{p^{D \setminus \{s\}}(S \setminus \{s\})}{1 - p(s) - \sum_{s' \in S \setminus \{s\}} p(s')} - \frac{p(S \setminus \{s\})}{1 - \sum_{s' \in S \setminus \{s\}} p(s')}$$

$$= \frac{p^{D \setminus \{s\}}(S \setminus \{s\})}{1 - \sum_{s' \in S} p(s')} - \frac{p(S \setminus \{s\})}{1 - \sum_{s' \in S \setminus \{s\}} p(s')}.$$

Multiplying by $1 - \sum_{s' \in S} p(s')$ and rearranging terms proves Lemma 2. $\qquad\square$

**Lemma 3.**

$$p(s) + \left( 1 - \sum_{s' \in S^k} p(s') \right) P(b_k = s | S^k) = p(s) R(S^k, s) \tag{39}$$

*Proof.* First using Lemma 1 and then Lemma 2 we find

$$p(s) + \left( 1 - \sum_{s' \in S^k} p(s') \right) P(b_k = s | S^k)$$

$$= p(s) + \left( 1 - \sum_{s' \in S^k} p(s') \right) \frac{p(S^k \setminus \{s\})}{p(S^k)} \frac{p(s)}{1 - \sum_{s' \in S^k \setminus \{s\}} p(s')}$$

$$= \frac{p(s)}{p(S^k)} \left( p(S^k) + \frac{1 - \sum_{s' \in S^k} p(s')}{1 - \sum_{s' \in S^k \setminus \{s\}} p(s')} p(S^k \setminus \{s\}) \right)$$

$$= \frac{p(s)}{p(S^k)} p^{D \setminus \{s\}}(S^k \setminus \{s\})$$

$$= p(s) R(S^k, s).$$

$\qquad\square$

Now we can complete the proof of Theorem 4 by adding $p(b_k)f(b_k) - p(b_k)f(b_k) = 0$ to the estimator, moving the terms independent of $B^k$ outside the expectation and using Lemma 3:

$$\mathbb{E}_{B^k \sim p(B^k|S^k)} \left[ \sum_{j=1}^{k-1} p(b_j)f(b_j) + \left( 1 - \sum_{j=1}^{k-1} p(b_j) \right) f(b_k) \right]$$

$$= \mathbb{E}_{B^k \sim p(B^k|S^k)} \left[ \sum_{j=1}^{k} p(b_j)f(b_j) + \left( 1 - \sum_{j=1}^{k} p(b_j) \right) f(b_k) \right]$$

$$= \sum_{s \in S^k} p(s)f(s) + \mathbb{E}_{B^k \sim p(B^k|S^k)} \left[ \left( 1 - \sum_{s' \in S^k} p(s') \right) f(b_k) \right]$$

$$= \sum_{s \in S^k} p(s)f(s) + \sum_{s \in S^k} \left( 1 - \sum_{s' \in S^k} p(s') \right) P(b_k = s|S^k)f(s)$$

$$= \sum_{s \in S^k} \left( p(s) + \left( 1 - \sum_{s' \in S^k} p(s') \right) P(b_k = s|S^k) \right) f(s)$$

$$= \sum_{s \in S^k} p(s)R(S^k, s)f(s).$$

$\square$

## C.3 THE STOCHASTIC SUM-AND-SAMPLE ESTIMATOR WITH MULTIPLE SAMPLES

As was discussed in Liu et al. (2019), one can trade off the number of summed terms and number of sampled terms to maximize the achieved variance reduction. As a generalization of Theorem 4 (the stochastic sum-and-sample estimator with $k - 1$ summed terms), we introduce here the stochastic sum-and-sample estimator that sums $k - m$ terms and samples $m > 1$ terms *without replacement*. To estimate the sampled term, we use the unordered set estimator on the $m$ samples without replacement, on the domain restricted to $D \setminus B^{k-m}$. In general, we denote the *unordered set estimator* restricted to the domain $D \setminus C$ by

$$e^{\text{US},D\setminus C}(S^k) = \sum_{s \in S^k \setminus C} p(s)R^{D\setminus C}(S^k, s)f(s) \tag{40}$$

where $R^{D\setminus C}(S^k, s)$ is the *leave-one-out ratio* restricted to the domain $D \setminus C$, similar to the second order leave-one-out ratio in equation 18:

$$R^{D\setminus C}(S^k, s) = \frac{p_{\boldsymbol{\theta}}^{(D\setminus C)\setminus\{s\}}((S^k \setminus C) \setminus \{s\})}{p_{\boldsymbol{\theta}}^{D\setminus C}(S^k \setminus C)}. \tag{41}$$

While we can also constrain $S^k \subseteq (D \setminus C)$, this definition is consistent with equation 18 and allows simplified notation.

**Theorem 5.** *Rao-Blackwellizing the stochastic sum-and-sample estimator with $m > 1$ samples results in the unordered set estimator, i.e.*

$$\mathbb{E}_{B^k \sim p(B^k|S^k)} \left[ \sum_{j=1}^{k-m} p(b_j)f(b_j) + \left( 1 - \sum_{j=1}^{k-m} p(b_j) \right) e^{US,D\setminus B^{k-m}}(S^k) \right] = \sum_{s \in S^k} p(s)R(S^k, s)f(s). \tag{42}$$

*Proof.* Recall that for the unordered set estimator, it holds that

$$e^{\text{US}}(S^k) = \mathbb{E}_{b_1 \sim p(b_1|S^k)} [f(b_1)] = \mathbb{E}_{x \sim p(x)} \left[ f(x) \big| x \in S^k \right] \tag{43}$$

$\square$

which for the restricted equivalent (with restricted distribution $p^{D \setminus C}$) translates into

$$e^{\text{US},D \setminus C}(S^k) = \mathbb{E}_{x \sim p^{D \setminus C}(x)}\left[f(x)\big|x \in S^k\right] = \mathbb{E}_{x \sim p(x)}\left[f(x)\big|x \in S^k, x \notin C\right]. \qquad (44)$$

Now we consider the distribution $b_{k-m+1}|S^k, B^{k-m}$: the distribution of the first element sampled (without replacement) after sampling $B^{k-m}$, given (conditionally on the event) that the set of $k$ samples is $S^k$, so we have $b_{k-m+1} \in S^k$ and $b_{k-m+1} \notin B^{k-m}$. This means that its conditional expectation of $f(b_{k-m+1})$ is the restricted unordered set estimator for $C = B^{k-m}$ since

$$\begin{aligned} e^{\text{US},D \setminus B^{k-m}}(S^k) &= \mathbb{E}_{x \sim p(x)}\left[f(x)\big|x \in S^k, x \notin B^{k-m}\right] \\ &= \mathbb{E}_{b_{k-m+1} \sim p(b_{k-m+1}|S^k, B^{k-m})}\left[f(b_{k-m+1})\right]. \end{aligned} \qquad (45)$$

Observing that the definition (equation 42) of the stochastic sum-and-sample estimator does not depend on the actual order of the $m$ samples, and using equation 45, we can reduce the multi-sample estimator to the stochastic sum-and-sample estimator with $k' = k - m + 1$, such that the result follows from equation 36.

$$\begin{aligned} &\mathbb{E}_{B^k \sim p(B^k|S^k)}\left[\sum_{j=1}^{k-m}p(b_j)f(b_j) + \left(1 - \sum_{j=1}^{k-m}p(b_j)\right)e^{\text{US},D \setminus B^{k-m}}(S^k)\right] \\ =&\mathbb{E}_{B^{k-m} \sim p(B^{k-m}|S^k)}\left[\sum_{j=1}^{k-m}p(b_j)f(b_j) + \left(1 - \sum_{j=1}^{k-m}p(b_j)\right)e^{\text{US},D \setminus B^{k-m}}(S^k)\right] \\ =&\mathbb{E}_{B^{k-m} \sim p(B^{k-m}|S^k)}\left[\sum_{j=1}^{k-m}p(b_j)f(b_j) + \left(1 - \sum_{j=1}^{k-m}p(b_j)\right)\mathbb{E}_{b_{k-m+1} \sim p(b_{k-m+1}|S^k, B^{k-m})}\left[f(b_{k-m+1})\right]\right] \\ =&\mathbb{E}_{B^{k-m+1} \sim p(B^{k-m+1}|S^k)}\left[\sum_{j=1}^{k-m}p(b_j)f(b_j) + \left(1 - \sum_{j=1}^{k-m}p(b_j)\right)f(b_{k-m+1})\right] \\ =&\mathbb{E}_{S^{k-m+1}|S^k}\left[\mathbb{E}_{B^{k-m+1} \sim p(B^{k-m+1}|S^{k-m+1})}\left[\sum_{j=1}^{k-m}p(b_j)f(b_j) + \left(1 - \sum_{j=1}^{k-m}p(b_j)\right)f(b_{k-m+1})\right]\right] \\ =&\mathbb{E}_{S^{k-m+1}|S^k}\left[\sum_{s \in S^k}p(s)R(S^k,s)f(s)\right] \\ =&\sum_{s \in S^k}p(s)R(S^k,s)f(s). \end{aligned} \qquad (46)$$

## D   THE IMPORTANCE-WEIGHTED ESTIMATOR

### D.1   RAO-BLACKWELLIZATION OF THE IMPORTANCE-WEIGHTED ESTIMATOR

In this section we give the proof that Rao-Blackwellizing the importance-weighted estimator results in the unordered set estimator.

**Theorem 6.** *Rao-Blackwellizing the importance-weighted estimator results in the unordered set estimator, i.e.:*

$$\mathbb{E}_{\kappa \sim p(\kappa|S^k)}\left[\sum_{s \in S^k}\frac{p(s)}{1 - F_{\phi_s}(\kappa)}f(s)\right] = \sum_{s \in S^k}p(s)R(S^k,s)f(s). \qquad (47)$$

Here we have slightly rewritten the definition of the importance-weighted estimator, using that $q(s,a) = P(g_{\phi_s} > a) = 1 - F_{\phi_s}(a)$, where $F_{\phi_s}$ is the CDF of the Gumbel distribution (see Appendix A).

*Proof.* We first prove the following Lemma:

**Lemma 4.**

$$\mathbb{E}_{\kappa \sim p(\kappa|S^k)}\left[\frac{1}{1-F_{\phi_s}(\kappa)}\right] = R(S^k, s) \tag{48}$$

*Proof.* Conditioning on $S^k$, we know that the elements in $S^k$ have the $k$ largest perturbed log-probabilities, so $\kappa$, the $(k+1)$-th largest perturbed log-probability is the largest perturbed log-probability in $D \setminus S^k$, and satisfies $\kappa = \max_{s \in D \setminus S^k} g_{\phi_s} = g_{\phi_{D \setminus S^k}} \sim \text{Gumbel}(\phi_{D \setminus S^k})$. Computing $p(\kappa|S^k)$ using Bayes' Theorem, we have

$$p(\kappa|S^k) = \frac{p(S^k|\kappa)p(\kappa)}{p(S^k)} = \frac{\prod_{s \in S^k}(1-F_{\phi_s}(\kappa))f_{\phi_{D \setminus S^k}}(\kappa)}{p(S^k)} \tag{49}$$

which allows us to compute (using equation 34 with $C = \{s\}$ and $g_{\phi_{D \setminus S}} = \kappa$)

$$\mathbb{E}_{\kappa \sim p(\kappa|S^k)}\left[\frac{1}{1-F_{\phi_s}(\kappa)}\right]$$
$$= \int_{\kappa=-\infty}^{\infty} p(\kappa|S^k)\frac{1}{1-F_{\phi_s}(\kappa)}\partial\kappa$$
$$= \int_{\kappa=-\infty}^{\infty} \frac{\prod_{s \in S^k}(1-F_{\phi_s}(\kappa))f_{\phi_{D \setminus S^k}}(\kappa)}{p(S^k)}\frac{1}{1-F_{\phi_s}(\kappa)}\partial\kappa$$
$$= \frac{1}{p(S^k)}\int_{\kappa=-\infty}^{\infty} \prod_{s \in S^k \setminus \{s\}}(1-F_{\phi_s}(\kappa))f_{\phi_{D \setminus S^k}}(\kappa)\partial\kappa$$
$$= \frac{1}{p(S^k)}p^{D \setminus \{s\}}(S \setminus \{s\})$$
$$= R(S^k, s).$$

$\square$

Using Lemma 4 we find

$$\mathbb{E}_{\kappa \sim p(\kappa|S^k)}\left[\sum_{s \in S^k}\frac{p(s)}{1-F_{\phi_s}(\kappa)}f(s)\right]$$
$$= \sum_{s \in S^k} p(s)\mathbb{E}_{\kappa \sim p(\kappa|S^k)}\left[\frac{1}{1-F_{\phi_s}(\kappa)}\right]f(s)$$
$$= \sum_{s \in S^k} p(s)R(S^k, s)f(s).$$

$\square$

## D.2 THE IMPORTANCE-WEIGHTED POLICY GRADIENT ESTIMATOR WITH BUILT-IN BASELINE

For self-containment we include this section, which is adapted from our unpublished workshop paper (Kool et al., 2019b). The importance-weighted policy gradient estimator combines REIN-FORCE (Williams, 1992) with the importance-weighted estimator (Duffield et al., 2007; Vieira, 2017) in equation 15 which results in an unbiased estimator of the policy gradient $\nabla_{\boldsymbol{\theta}}\mathbb{E}_{p_{\boldsymbol{\theta}}(x)}[f_{\boldsymbol{\theta}}(x)]$:

$$e^{\text{IWPG}}(S^k, \kappa) = \sum_{s \in S^k}\frac{p_{\boldsymbol{\theta}}(s)}{q_{\boldsymbol{\theta},\kappa}(s)}\nabla_{\boldsymbol{\theta}}\log p_{\boldsymbol{\theta}}(s)f(s) = \sum_{s \in S^k}\frac{\nabla_{\boldsymbol{\theta}}p_{\boldsymbol{\theta}}(s)}{q_{\boldsymbol{\theta},\kappa}(s)}f(s) \tag{50}$$

Recall that $\kappa$ is the $(k+1)$-th largest perturbed log-probability (see Section 3.2). We compute a lower variance but biased variant by normalizing the importance weights using the normalization $W(S^k) = \sum_{s \in S^k}\frac{p_{\boldsymbol{\theta}}(s)}{q_{\boldsymbol{\theta},\kappa}(s)}$.

As we show in Kool et al. (2019b), we can include a 'baseline' $B(S^k) = \sum_{s \in S^k} \frac{p_{\boldsymbol{\theta}}(s)}{q_{\boldsymbol{\theta},\kappa}(s)} f(s)$ and correct for the bias (since it depends on the complete sample $S^k$) by weighting individual terms of the estimator by $1 - p_{\boldsymbol{\theta}}(s) + \frac{p_{\boldsymbol{\theta}}(s)}{q_{\boldsymbol{\theta},\kappa}(s)}$:

$$e^{\text{IWPGBL}}(S^k, \kappa) = \sum_{s \in S^k} \frac{\nabla_{\boldsymbol{\theta}} p_{\boldsymbol{\theta}}(s)}{q_{\boldsymbol{\theta},\kappa}(s)} \left( f(s) \left( 1 - p_{\boldsymbol{\theta}}(s) + \frac{p_{\boldsymbol{\theta}}(s)}{q_{\boldsymbol{\theta},\kappa}(s)} \right) - B(S^k) \right) \tag{51}$$

For the normalized version, we use the normalization $W(S^k) = \sum_{s \in S^k} \frac{p_{\boldsymbol{\theta}}(s)}{q_{\boldsymbol{\theta},\kappa}(s)}$ for the baseline, and $W_i(S^k) = W(S^k) - \frac{p_{\boldsymbol{\theta}}(s)}{q_{\boldsymbol{\theta},\kappa}(s)} + p_{\boldsymbol{\theta}}(s)$ to normalize the individual terms:

$$\nabla_{\boldsymbol{\theta}} \mathbb{E}_{y \sim p_{\boldsymbol{\theta}}(y)}[f(y)] \approx \sum_{s \in S^k} \frac{1}{W_i(S^k)} \cdot \frac{\nabla_{\boldsymbol{\theta}} p_{\boldsymbol{\theta}}(s)}{q_{\boldsymbol{\theta},\kappa}(s)} \left( f(s) - \frac{B(S^k)}{W(S^k)} \right) \tag{52}$$

It seems odd to normalize the terms in the outer sum by $\frac{1}{W_i(S^k)}$ instead of $\frac{1}{W(S^k)}$, but equation 52 can be rewritten into a form similar to equation 17, i.e. with a different baseline for each sample, but this form is more convenient for implementation (Kool et al., 2019b).

## E    THE UNORDERED SET POLICY GRADIENT ESTIMATOR

### E.1    PROOF OF UNBIASEDNESS OF THE UNORDERED SET POLICY GRADIENT ESTIMATOR WITH BASELINE

To prove the unbiasedness of result we need to prove that the control variate has expectation 0:

**Lemma 5.**

$$\mathbb{E}_{S^k \sim p_{\boldsymbol{\theta}}(S^k)} \left[ \sum_{s \in S^k} \nabla_{\boldsymbol{\theta}} p_{\boldsymbol{\theta}}(s) R(S^k, s) \sum_{s' \in S^k} p_{\boldsymbol{\theta}}(s') R^{D \setminus \{s\}}(S^k, s') f(s') \right] = 0. \tag{53}$$

*Proof.* Similar to equation 10, we apply Bayes' Theorem conditionally on $b_1 = s$ to derive for $s' \neq s$

$$\begin{aligned} P(b_2 = s' | S^k, b_1 = s) &= \frac{P(S^k | b_2 = s', b_1 = s) P(b_2 = s' | b_1 = s')}{P(S^k | b_1 = s)} \\ &= \frac{p_{\boldsymbol{\theta}}^{D \setminus \{s, s'\}}(S^k \setminus \{s, s'\}) p_{\boldsymbol{\theta}}^{D \setminus \{s\}}(s')}{p_{\boldsymbol{\theta}}^{D \setminus \{s\}}(S^k \setminus \{s\})} \\ &= \frac{p_{\boldsymbol{\theta}}(s')}{1 - p_{\boldsymbol{\theta}}(s)} R^{D \setminus \{s\}}(S^k, s'). \end{aligned} \tag{54}$$

For $s' = s$ we have $R^{D \setminus \{s\}}(S^k, s') = 1$ by definition, so using equation 54 we can show that

$$\begin{aligned} &\sum_{s' \in S^k} p_{\boldsymbol{\theta}}(s') R^{D \setminus \{s\}}(S^k, s') f(s') \\ &= p_{\boldsymbol{\theta}}(s) f(s) + \sum_{s' \in S^k \setminus \{s\}} p_{\boldsymbol{\theta}}(s') R^{D \setminus \{s\}}(S^k, s') f(s') \\ &= p_{\boldsymbol{\theta}}(s) f(s) + (1 - p_{\boldsymbol{\theta}}(s)) \sum_{s' \in S^k \setminus \{s\}} \frac{p_{\boldsymbol{\theta}}(s')}{1 - p_{\boldsymbol{\theta}}(s)} R^{D \setminus \{s\}}(S^k, s') f(s') \\ &= p_{\boldsymbol{\theta}}(s) f(s) + (1 - p_{\boldsymbol{\theta}}(s)) \sum_{s' \in S^k \setminus \{s\}} P(b_2 = s' | S^k, b_1 = s) f(s') \\ &= p_{\boldsymbol{\theta}}(s) f(s) + (1 - p_{\boldsymbol{\theta}}(s)) \mathbb{E}_{b_2 \sim p_{\boldsymbol{\theta}}(b_2 | S^k, b_1 = s)}[f(b_2)] \\ &= \mathbb{E}_{b_2 \sim p_{\boldsymbol{\theta}}(b_2 | S^k, b_1 = s)}[p_{\boldsymbol{\theta}}(b_1) f(b_1) + (1 - p_{\boldsymbol{\theta}}(b_1)) f(b_2)]. \end{aligned}$$

Now we can show that the control variate is actually the result of Rao-Blackwellization:

$$\mathbb{E}_{S^k \sim p_{\boldsymbol{\theta}}(S^k)} \left[ \sum_{s \in S^k} \nabla_{\boldsymbol{\theta}} p_{\boldsymbol{\theta}}(s) R(S^k, s) \sum_{s' \in S^k} p_{\boldsymbol{\theta}}(s') R^{D \setminus \{s\}}(S^k, s') f(s') \right]$$

$$= \mathbb{E}_{S^k \sim p_{\boldsymbol{\theta}}(S^k)} \left[ \sum_{s \in S^k} p_{\boldsymbol{\theta}}(s) R(S^k, s) \nabla_{\boldsymbol{\theta}} \log p_{\boldsymbol{\theta}}(s) \sum_{s' \in S^k} p_{\boldsymbol{\theta}}(s') R^{D \setminus \{s\}}(S^k, s') f(s') \right]$$

$$= \mathbb{E}_{S^k \sim p_{\boldsymbol{\theta}}(S^k)} \left[ \sum_{s \in S^k} P(b_1 = s | S^k) \nabla_{\boldsymbol{\theta}} \log p_{\boldsymbol{\theta}}(s) \sum_{s' \in S^k} p_{\boldsymbol{\theta}}(s') R^{D \setminus \{s\}}(S^k, s') f(s') \right]$$

$$= \mathbb{E}_{S^k \sim p_{\boldsymbol{\theta}}(S^k)} \left[ \mathbb{E}_{b_1 \sim p_{\boldsymbol{\theta}}(b_1 | S^k)} \left[ \nabla_{\boldsymbol{\theta}} \log p_{\boldsymbol{\theta}}(b_1) \sum_{s' \in S^k} p_{\boldsymbol{\theta}}(s') R^{D \setminus \{b_1\}}(S^k, s') f(s') \right] \right]$$

$$= \mathbb{E}_{S^k \sim p_{\boldsymbol{\theta}}(S^k)} \left[ \mathbb{E}_{b_1 \sim p_{\boldsymbol{\theta}}(b_1 | S^k)} \left[ \nabla_{\boldsymbol{\theta}} \log p_{\boldsymbol{\theta}}(b_1) \mathbb{E}_{b_2 \sim p_{\boldsymbol{\theta}}(b_2 | S^k, b_1)} \left[ p_{\boldsymbol{\theta}}(b_1) f(b_1) + (1 - p_{\boldsymbol{\theta}}(b_1)) f(b_2) \right] \right] \right]$$

$$= \mathbb{E}_{S^k \sim p_{\boldsymbol{\theta}}(S^k)} \left[ \mathbb{E}_{B^k \sim p_{\boldsymbol{\theta}}(B^k | S^k)} \left[ \nabla_{\boldsymbol{\theta}} \log p_{\boldsymbol{\theta}}(b_1) \left( p_{\boldsymbol{\theta}}(b_1) f(b_1) + (1 - p_{\boldsymbol{\theta}}(b_1)) f(b_2) \right) \right] \right]$$

$$= \mathbb{E}_{B^k \sim p_{\boldsymbol{\theta}}(B^k)} \left[ \nabla_{\boldsymbol{\theta}} \log p_{\boldsymbol{\theta}}(b_1) \left( p_{\boldsymbol{\theta}}(b_1) f(b_1) + (1 - p_{\boldsymbol{\theta}}(b_1)) f(b_2) \right) \right]$$

This expression depends only on $b_1$ and $b_2$ and we recognize the stochastic sum-and-sample estimator for $k = 2$ used as 'baseline'. As a special case of equation 13 for $C = \{b_1\}$, we have

$$\mathbb{E}_{b_2 \sim p_{\boldsymbol{\theta}}(b_2 | b_1)} \left[ (p_{\boldsymbol{\theta}}(b_1) f(b_1) + (1 - p_{\boldsymbol{\theta}}(b_1)) f(b_2)) \right] = \mathbb{E}_{i \sim p_{\boldsymbol{\theta}}(i)} \left[ f(i) \right]. \tag{55}$$

Using this, and the fact that $\mathbb{E}_{b_1 \sim p_{\boldsymbol{\theta}}(b_1)} \left[ \nabla_{\boldsymbol{\theta}} \log p_{\boldsymbol{\theta}}(b_1) \right] = \nabla_{\boldsymbol{\theta}} \mathbb{E}_{b_1 \sim p_{\boldsymbol{\theta}}(b_1)} \left[ 1 \right] = \nabla_{\boldsymbol{\theta}} 1 = 0$ we find

$$\mathbb{E}_{S^k \sim p_{\boldsymbol{\theta}}(S^k)} \left[ \sum_{s \in S^k} \nabla_{\boldsymbol{\theta}} p_{\boldsymbol{\theta}}(s) R(S^k, s) \sum_{s' \in S^k} p_{\boldsymbol{\theta}}(s') R^{D \setminus \{s\}}(S^k, s') f(s') \right]$$

$$= \mathbb{E}_{B^k \sim p_{\boldsymbol{\theta}}(B^k)} \left[ \nabla_{\boldsymbol{\theta}} \log p_{\boldsymbol{\theta}}(b_1) \left( p_{\boldsymbol{\theta}}(b_1) f(b_1) + (1 - p_{\boldsymbol{\theta}}(b_1)) f(b_2) \right) \right]$$

$$= \mathbb{E}_{b_1 \sim p_{\boldsymbol{\theta}}(b_1)} \left[ \nabla_{\boldsymbol{\theta}} \log p_{\boldsymbol{\theta}}(b_1) \mathbb{E}_{b_2 \sim p_{\boldsymbol{\theta}}(b_2 | b_1)} \left[ (p_{\boldsymbol{\theta}}(b_1) f(b_1) + (1 - p_{\boldsymbol{\theta}}(b_1)) f(b_2)) \right] \right]$$

$$= \mathbb{E}_{b_1 \sim p_{\boldsymbol{\theta}}(b_1)} \left[ \nabla_{\boldsymbol{\theta}} \log p_{\boldsymbol{\theta}}(b_1) \mathbb{E}_{x \sim p_{\boldsymbol{\theta}}(x)} \left[ f(x) \right] \right]$$

$$= \mathbb{E}_{b_1 \sim p_{\boldsymbol{\theta}}(b_1)} \left[ \nabla_{\boldsymbol{\theta}} \log p_{\boldsymbol{\theta}}(b_1) \right] \mathbb{E}_{x \sim p_{\boldsymbol{\theta}}(x)} \left[ f(x) \right]$$

$$= 0 \cdot \mathbb{E}_{x \sim p_{\boldsymbol{\theta}}(x)} \left[ f(x) \right]$$

$$= 0$$

$$\square$$

## F  THE RISK ESTIMATOR

### F.1  PROOF OF BUILT-IN BASELINE

We show that the RISK estimator, taking gradients through the normalization factor actually has a built-in baseline. We first use the log-derivative trick to rewrite the gradient of the ratio as the ratio times the logarithm of the gradient, and then swap the summation variables in the double sum that arises:

$$
\begin{aligned}
e^{\text{RISK}}(S) &= \sum_{s \in S} \nabla_{\boldsymbol{\theta}} \left( \frac{p_{\boldsymbol{\theta}}(s)}{\sum_{s' \in S} p_{\boldsymbol{\theta}}(s')} \right) f(s) \\
&= \sum_{s \in S} \frac{p_{\boldsymbol{\theta}}(s)}{\sum_{s' \in S} p_{\boldsymbol{\theta}}(s')} \nabla_{\boldsymbol{\theta}} \log \left( \frac{p_{\boldsymbol{\theta}}(s)}{\sum_{s' \in S} p_{\boldsymbol{\theta}}(s')} \right) f(s) \\
&= \sum_{s \in S} \frac{p_{\boldsymbol{\theta}}(s)}{\sum_{s' \in S} p_{\boldsymbol{\theta}}(s')} \left( \nabla_{\boldsymbol{\theta}} \log p_{\boldsymbol{\theta}}(s) - \nabla_{\boldsymbol{\theta}} \log \sum_{s' \in S} p_{\boldsymbol{\theta}}(s') \right) f(s) \\
&= \sum_{s \in S} \frac{p_{\boldsymbol{\theta}}(s)}{\sum_{s' \in S} p_{\boldsymbol{\theta}}(s')} \left( \frac{\nabla_{\boldsymbol{\theta}} p_{\boldsymbol{\theta}}(s)}{p_{\boldsymbol{\theta}}(s)} - \frac{\sum_{s' \in S} \nabla_{\boldsymbol{\theta}} p_{\boldsymbol{\theta}}(s')}{\sum_{s' \in S} p_{\boldsymbol{\theta}}(s')} \right) f(s) \\
&= \sum_{s \in S} \frac{\nabla_{\boldsymbol{\theta}} p_{\boldsymbol{\theta}}(s) f(s)}{\sum_{s' \in S} p_{\boldsymbol{\theta}}(s')} - \frac{\sum_{s,s' \in S} p_{\boldsymbol{\theta}}(s) \nabla_{\boldsymbol{\theta}} p_{\boldsymbol{\theta}}(s') f(s)}{\left( \sum_{s' \in S} p_{\boldsymbol{\theta}}(s') \right)^2} \\
&= \sum_{s \in S} \frac{\nabla_{\boldsymbol{\theta}} p_{\boldsymbol{\theta}}(s) f(s)}{\sum_{s' \in S} p_{\boldsymbol{\theta}}(s')} - \frac{\sum_{s,s' \in S} p_{\boldsymbol{\theta}}(s') \nabla_{\boldsymbol{\theta}} p_{\boldsymbol{\theta}}(s) f(s')}{\left( \sum_{s' \in S} p_{\boldsymbol{\theta}}(s') \right)^2} \\
&= \sum_{s \in S} \frac{\nabla_{\boldsymbol{\theta}} p_{\boldsymbol{\theta}}(s)}{\sum_{s' \in S} p_{\boldsymbol{\theta}}(s')} \left( f(s) - \frac{\sum_{s' \in S} p_{\boldsymbol{\theta}}(s') f(s')}{\sum_{s' \in S} p_{\boldsymbol{\theta}}(s')} \right) \\
&= \sum_{s \in S} \frac{\nabla_{\boldsymbol{\theta}} p_{\boldsymbol{\theta}}(s)}{\sum_{s'' \in S} p_{\boldsymbol{\theta}}(s'')} \left( f(s) - \sum_{s' \in S} \frac{p_{\boldsymbol{\theta}}(s')}{\sum_{s'' \in S} p_{\boldsymbol{\theta}}(s'')} f(s') \right).
\end{aligned}
$$

## G  CATEGORICAL VARIATIONAL AUTO-ENCODER

### G.1  EXPERIMENTAL DETAILS

We use the code[6] by Yin et al. (2019) to reproduce their categorical VAE experiment, of which we include details here for self-containment. The dataset is MNIST, statically binarized by thresholding at 0.5 (although we include results using the standard binarized dataset by Salakhutdinov & Murray (2008); Larochelle & Murray (2011) in Section G.2). The latent representation $\boldsymbol{z}$ is $K = 20$ dimensional with $C = 10$ categories per dimension with a uniform prior $p(z_k = c) = 1/C, k = 1, ..., K$. The encoder is parameterized by $\phi$ as $q_{\phi}(\boldsymbol{z}|\boldsymbol{x}) = \prod_k q_{\phi}(z_k|\boldsymbol{x})$ and has two fully connected hidden layers with 512 and 256 hidden nodes respectively, with LeakyReLU ($\alpha = 0.1$) activations. The decoder, parameterized by $\boldsymbol{\theta}$, is given by $p_{\boldsymbol{\theta}}(\boldsymbol{x}|\boldsymbol{z}) = \prod_i p_{\boldsymbol{\theta}}(x_i|\boldsymbol{z})$, where $x_i \in \{0, 1\}$ are the pixel values, and has fully connected hidden layers with 256 and 512 nodes and LeakyReLU activation.

**ELBO optimization.**  The evidence lower bound (ELBO) that we optimize is given by

$$
\mathcal{L}(\phi, \boldsymbol{\theta}) = \mathbb{E}_{\boldsymbol{z} \sim q_{\phi}(\boldsymbol{z}|\boldsymbol{x})} \left[ \ln p_{\boldsymbol{\theta}}(\boldsymbol{x}|\boldsymbol{z}) + \ln p(\boldsymbol{z}) - \ln q_{\phi}(\boldsymbol{z}|\boldsymbol{x}) \right] \tag{56}
$$

$$
= \mathbb{E}_{\boldsymbol{z} \sim q_{\phi}(\boldsymbol{z}|\boldsymbol{x})} \left[ \ln p_{\boldsymbol{\theta}}(\boldsymbol{x}|\boldsymbol{z}) \right] - KL(q_{\phi}(\boldsymbol{z}|\boldsymbol{x}) || p(\boldsymbol{z})) . \tag{57}
$$

For the decoder parameters $\boldsymbol{\theta}$, since $q_{\phi}(\boldsymbol{z}|\boldsymbol{x})$ does not depend on $\boldsymbol{\theta}$, it follows that

$$
\nabla_{\boldsymbol{\theta}} \mathcal{L}(\phi, \boldsymbol{\theta}) = \mathbb{E}_{\boldsymbol{z} \sim q_{\phi}(\boldsymbol{z}|\boldsymbol{x})} \left[ \nabla_{\boldsymbol{\theta}} \ln p_{\boldsymbol{\theta}}(\boldsymbol{x}|\boldsymbol{z}) \right] . \tag{58}
$$

For the encoder parameters $\phi$, we can write $\nabla_{\phi} \mathcal{L}(\phi, \boldsymbol{\theta})$ using equation 57 and equation 19 as

$$
\nabla_{\phi} \mathcal{L}(\phi, \boldsymbol{\theta}) = \mathbb{E}_{\boldsymbol{z} \sim q_{\phi}(\boldsymbol{z}|\boldsymbol{x})} \left[ \nabla_{\phi} \ln q_{\phi}(\boldsymbol{z}|\boldsymbol{x}) \ln p_{\boldsymbol{\theta}}(\boldsymbol{x}|\boldsymbol{z}) \right] - \nabla_{\phi} KL(q_{\phi}(\boldsymbol{z}|\boldsymbol{x}) || p(\boldsymbol{z})) . \tag{59}
$$

---

[6]https://github.com/ARM-gradient/ARSM

This assumes we can compute the KL divergence analytically. Alternatively, we can use a sample estimate for the KL divergence, and use equation 56 with equation 19 to obtain

$$\nabla_\phi \mathcal{L}(\phi, \theta) = \mathbb{E}_{z \sim q_\phi(z|x)} \left[ \nabla_\phi \ln q_\phi(z|x)(\ln p_\theta(x|z) + \ln p(z) - \ln q_\phi(z|x)) + \nabla_\phi \ln q_\phi(z|x) \right] \tag{60}$$

$$= \mathbb{E}_{z \sim q_\phi(z|x)} \left[ \nabla_\phi \ln q_\phi(z|x)(\ln p_\theta(x|z) - \ln q_\phi(z|x)) \right]. \tag{61}$$

Here we have left out the term $\mathbb{E}_{z \sim q_\phi(z|x)} \left[ \nabla_\phi \ln q_\phi(z|x) \right] = 0$, similar to Roeder et al. (2017), and, assuming a uniform (i.e. constant) prior $\ln p(z)$, the term $\mathbb{E}_{z \sim q_\phi(z|x)} \left[ \nabla_\phi \ln q_\phi(z|x) \ln p(z) \right] = 0$. With a built-in baseline, this second term cancels out automatically, even if it is implemented. Despite the similarity of the equation 56 and equation 57, their gradient estimates (equation 60 and equation 59) are structurally dissimilar and care should be taken to implement the REINFORCE estimator (or related estimators such as ARSM and the unordered set estimator) correctly using automatic differentiation software. Using Gumbel-Softmax and RELAX, we take gradients 'directly' through the objective in equation 57.

We optimize the ELBO using the analytic KL for 1000 epochs using the Adam (Kingma & Ba, 2015) optimizer. We use a learning rate of $10^{-3}$ for all estimators except Gumbel-Softmax and RELAX, which use a learning rate of $10^{-4}$ as we found they diverged with a higher learning rate. For ARSM, as an exception we use the sample KL, and a learning rate of $3 \cdot 10^{-4}$, as suggested by the authors. All reported ELBO values are computed using the analytic KL. Our code is publicly available[7].

### G.2 Additional results

**Gradient variance during training.** We also evaluate gradient variance of different estimators during different stages of training. We measure the variance of different estimators with $k = 4$ samples during training with REINFORCE with replacement, such that all estimators are computed for the same model parameters. The results during training, given in Figure 4, are similar to the results for the trained model in Table 1, except for at the beginning of training, although the rankings of different estimator are mostly the same.

**Negative ELBO on validation set.** Figure 5 shows the -ELBO evaluated during training on the validation set. For the large latent space, we see validation error quickly increase (after reaching a minimum) which is likely because of overfitting (due to improved optimization), a phenomenon observed before (Tucker et al., 2017; Grathwohl et al., 2018). Note that before the overfitting starts, both REINFORCE without replacement and the unordered set estimator achieve a validation error similar to the other estimators, such that in a practical setting, one can use early stopping.

**Results using standard binarized MNIST dataset.** Instead of using the MNIST dataset binarized by thresholding values at 0.5 (as in the code and paper by Yin et al. (2019)) we also experiment with the standard (fixed) binarized dataset by Salakhutdinov & Murray (2008); Larochelle & Murray (2011), for which we plot train and validation curves for two runs on the small and large domain in Figure 6. This gives more realistic (higher) -ELBO scores, although we still observe the effect of overfitting. As this is a bit more unstable setting, one of the runs using REINFORCE with replacement diverged, but in general the relative performance of estimators is similar to using the dataset with 0.5 threshold.

---

[7]https://github.com/wouterkool/estimating-gradients-without-replacement

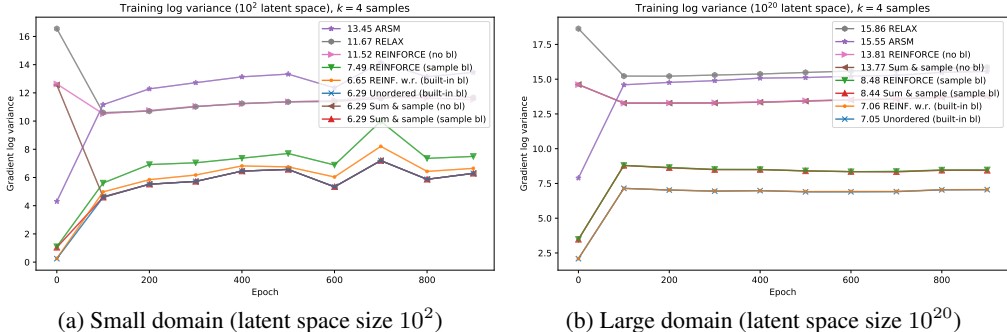

(a) Small domain (latent space size $10^2$)          (b) Large domain (latent space size $10^{20}$)

Figure 4: Gradient log variance of different unbiased estimators with $k = 4$ samples, estimated every 100 (out of 1000) epochs while training using REINFORCE with replacement. Each estimator is computed 1000 times with different latent samples for a fixed minibatch (the first 100 records of training data). We report (the logarithm of) the sum of the variances per parameter (trace of the covariance matrix). Some lines coincide, so we sort the legend by the last measurement and report its value.

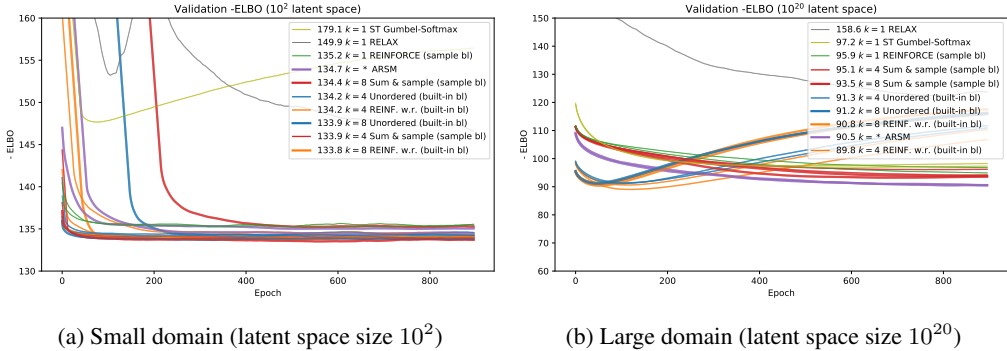

(a) Small domain (latent space size $10^2$)          (b) Large domain (latent space size $10^{20}$)

Figure 5: Smoothed validation -ELBO curves during training of two independent runs when with different estimators with $k = 1$, 4 or 8 (thicker lines) samples (ARSM has a variable number). Some lines coincide, so we sort the legend by the lowest -ELBO achieved and report this value.

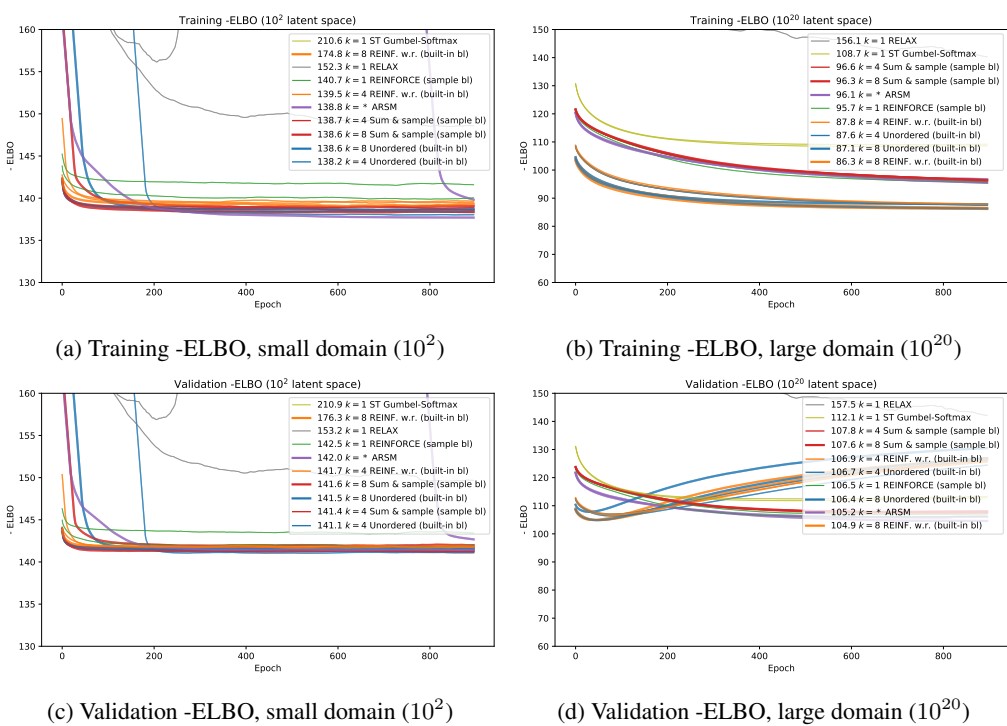

(a) Training -ELBO, small domain ($10^2$)

(b) Training -ELBO, large domain ($10^{20}$)

(c) Validation -ELBO, small domain ($10^2$)

(d) Validation -ELBO, large domain ($10^{20}$)

Figure 6: Smoothed training and validation -ELBO curves during training on the standard binarized MNIST dataset (Salakhutdinov & Murray, 2008; Larochelle & Murray, 2011) of two independent runs when with different estimators with $k = 1$, 4 or 8 (thicker lines) samples (ARSM has a variable number). Some lines coincide, so we sort the legend by the lowest -ELBO achieved and report this value.

## H  TRAVELLING SALESMAN PROBLEM

The Travelling Salesman Problem (TSP) is a discrete optimization problem that consists of finding the order in which to visit a set of locations, given as $x, y$ coordinates, to minimize the total length of the tour, starting and ending at the same location. As a tour can be considered a sequence of locations, this problem can be set up as a sequence modelling problem, that can be either addressed using supervised (Vinyals et al., 2015) or reinforcement learning (Bello et al., 2016; Kool et al., 2019a).

Kool et al. (2019a) introduced the Attention Model, which is an encoder-decoder model which considers a TSP instances as a fully connected graph. The encoder computes embeddings for all nodes (locations) and the decoder produces a tour, which is sequence of nodes, selecting one note at the time using an attention mechanism, and uses this autoregressively as input to select the next node. In Kool et al. (2019a), this model is trained using REINFORCE, with a greedy rollout used as baseline to reduce variance.

We use the code by Kool et al. (2019a) to train the exact same Attention Model (for details we refer to Kool et al. (2019a)), and minimize the expected length of a tour predicted by the model, using different gradient estimators. We did not do any hyperparameter optimization and used the exact same training details, using the Adam optimizer (Kingma & Ba, 2015) with a learning rate of $10^{-4}$ (no decay) for 100 epochs for all estimators. For the baselines, we used the same batch size of 512, but for estimators that use $k = 4$ samples, we used a batch size of $\frac{512}{4} = 128$ to compensate for the additional samples (this makes multi-sample methods actually faster since the encoder still needs to be evaluated only once).

