# OpenReview forum: "Estimating Gradients for Discrete Random Variables by Sampling without Replacement"
_ICLR.cc/2020/Conference — Accept (Spotlight)_

### Official Review · AnonReviewer3 · 2019-10-21
**Official Blind Review #3**

**Rating:** 8

**Review:**

Edited after rebuttal:
I'm satisfied with the clarifications provided by the authors about where they see this work being applicable. I think they have correctly identified a setting that is of interest to a broad audience and where their approach is more attractive than the alternatives. I now recommend accepting this paper.

The authors develop a generic method for estimating expectations under discrete distributions based on sampling without replacement and apply it to the task of estimating gradients for training models with discrete latent variables. The proposed estimator is unbiased and the authors prove it has lower variance than the naive Monte Carlo estimator.

I weakly recommend rejecting the paper. While the derivations and theorems presented in the paper are correct, the experiments are sensible and support the claims made, my impression is that the range of applications where the proposed method is a better alternative to the existing algorithms is extremely narrow. I would be willing to raise my rating to Weak Accept if the authors convince me that their method is useful more broadly than I think it is.

The method derivation is generally well-written and easy to follow. However, the account of the sum-and-sample estimator is somewhat misleading. As is, it suggests that it is natural to sum over k-1 elements and only draw one sample, while in fact choosing how many elements to sum over (I'll call this number S in my review) and how many samples to draw from the remaining set is the crucial step needed to make this algorithm work well. This is discussed by Fearnhead and Clifford [1] in the context of resampling in particle filters and I believe is also addressed by Liu et al. (2019). I understand that the authors are guiding the presentation towards a version of sum-and-sample they can compare with in theory and in experiments, but in the process they're doing a disservice to the reader. I think the authors should state the importance of choosing S well early on and then either compare with methods that attempt to set S optimally throughout experiments, or clearly explain that they restrict their discussion to a particular setting where S is difficult to optimize and explain why.

There are also additional baselines that should be considered. This includes methods such as stratified or systematic sampling, which are often discussed within the context of resampling [2]. Another strong but somewhat obscure baseline is given by Duffield et al. [3]. While it is possible that none of those methods are applicable in the experimental setting chosen in the paper, the authors should be very clear about how broadly they claim superiority of their method.

On the topic of experimental evaluation, the authors seem to have focused on the setting where sampling is done using stochastic beam search, presumably because in such a setting some of the strong baselines mentioned above, including sum-and-sample with the optimal choice of S, are not applicable. If that's true, I would like to see a detailed discussion about what it is about the problem setting that makes alternative baselines inapplicable. My current impression is that the proposed method is only worth considering when used with a stochastic beam search and I don't think such settings are particularly common. This is my main complaint about the paper.

On the subject of applicability, the experiments only test the proposed method in very low k settings, where the exponential algorithm for computing the leave-one-out ratio can be applied. While the appendix mentions that there is a faster alternative based on numerical integration, the claim that it can be efficiently integrated is not substantiated anywhere.

The plots could use some improvement. It took me a lot of squinting at Figures 2 and 3 to figure out what's going on, since the lines overlap too much - I couldn't even find the Unordered line in Figure 2 at first. Also I'm confused about why lines for sum-and-sample in Figure 1 extend beyond 8 evaluations, and VIMCO does not. Also figure captions could be a little more self-contained.

Finally, I would recommend stating Theorem 1 in more concrete terms, such as "estimator A has lower variance than estimator B".

[1] P. Fearnhead and P. Clifford. On-line inference for hidden Markov models via particle filters. Journal of the Royal Statistical Society, 65:887–899, 2003.
[2] R. Douc, O. Cappé, and E. Moulines. Comparison of resampling schemes for particle filtering. In ISPA, 2005.
[3] N. Duffield, C. Lund, and M. Thorup. Priority sampling for estimation of arbitrary subset sums. Journal of the ACM, 54, 2007.

**Experience Assessment:**

I have read many papers in this area.

**Review Assessment: Checking Correctness Of Derivations And Theory:**

I carefully checked the derivations and theory.

**Review Assessment: Checking Correctness Of Experiments:**

I carefully checked the experiments.

**Review Assessment: Thoroughness In Paper Reading:**

I read the paper thoroughly.

---

> ### Author Response · Authors · 2019-11-12
> **Detailed response to Reviewer 3**
>
> Thank you for your time reviewing our paper and for the feedback! We have used it to update the paper. Please see our detailed reply below:
>
> - Range of applications
> Given the popularity of Gumbel-Softmax, there is a wide interest in gradient estimation with discrete variables. However, a challenge remains to estimate gradients in high dimensional/structured settings, such as discrete representation learning, program synthesis, machine translation, image captioning and discrete generative modelling, where the Gumbel-Softmax bias accumulates due to mixing errors, and standard reinforce suffers from high variance (and supervised training is not possible or biased). Our estimator can be used as alternative to Gumbel-Softmax (and strong baselines such as VIMCO), while being more generally applicable in terms of domains (single dimensional and structured) and settings (high and low entropy). The theory, connecting various estimators, provides an important basis for developing further improved alternatives.
>
> - The sum-and-sample estimator
> Thank you, this is something that we did not realize as Liu et al. (2019) used a single sample in all experiments. We have included it in our discussion. Additionally, we have analyzed a version of the stochastic sum-and-sample estimator that sums k - m terms and takes m samples (without replacement). As it turns out, the Rao-Blackwellization result still holds even if the number of samples is chosen optimally (although the achieved variance reduction may be less). Regarding the experiments, choosing S optimally requires computation of model probabilities, which, in a structured setting (e.g. sequence model), is just as costly as taking a sample, so this is often not feasible, whereas our estimator does not require to select S.
>
> - Additional baselines
> We have added stratified and systematic sampling as baselines for the Bernoulli experiment and added a discussion on the limitations of these methods: it is unclear how strata should be defined in high-dimensional settings (such as a sequence model) and how to include a built-in baseline (since samples are no longer independent) when using stratified sampling with policy gradients. This built-in baseline (without adding bias) is a key component that we found makes our estimator work well in practice.
>
> Duffield et al. (2007) is what we refer to as the importance-weighted estimator, where we use ‘standard’ sampling without replacement (see https://timvieira.github.io/blog/post/2017/07/03/estimating-means-in-a-finite-universe/). While we found that this estimator without baseline is ineffective, we have derived a version with built-in baseline in an unpublished workshop paper, of which the normalized (biased) version is practically usable. We have included these results in the TSP experiment (with an anonymous reference that we will update after the review period). The unordered set estimator has better performance.
>
> We do not believe that more baselines would change the overall picture. We have tried to provide a fair evaluation of our estimator, including multiple strong baselines such as VIMCO and the (biased) risk estimator, while these have been often omitted in related work.
>
> - Experimental setting / focus on stochastic beam search
> Indeed, our focus is on applications where we can use stochastic beam search for sampling: there is a quickly growing body of work that considers learning with high-dimensional (structured) discrete output (or latent) spaces, such as machine translation, image captioning, generative modelling with structures (usually generated sequentially) such as molecules, program synthesis, discrete representation learning, etc. In such settings, Gumbel-Softmax suffers from mixing errors, and standard reinforce has high variance. For sum-and-sample estimators, it is not possible to obtain the ‘top k’ set (although it can be approximated), and non-trivial to sample from the remaining domain.
>
> - Applicability / numeric integration for large k
> In most cases, the low k setting is applicable because of diminishing returns for larger k. Numerical integration is feasible, although it requires setting a parameter a, for which we found a = 5 work well in practice. We have included details in the Appendix and an implementation (which computes all second order leave-one-out ratios efficiently) is provided in the code.
>
> - Plots improvement
> We have improved the caption of Figure 1: the sum-and-sample estimator uses *two* evaluations for every summed/sampled category as it also samples an independent baseline which is why the line extends twice as far (VIMCO has a built-in baseline). We have moved Figure 2 to the Appendix and replaced it by a table, making the results more clear.
>
> - Theorem 1
> The main implication of Theorem 1 is that the unordered set estimator is unbiased, as it is expected that using more than 1 sample reduces variance. We have clarified this in the paper.

---

> > ### Comment · AnonReviewer3 · 2019-11-15
> > **Concrete references for applications?**
> >
> > Your Rao-Blackwellization result holds for stochastic sum-and-sample with any choice of S, but I don't think it would hold if the elements in set C being summed over are also chosen optimally. In my experience sum-and-sample with optimal choice of C is very hard to beat and I don't think you're currently comparing to it in your experiments. This is fine if you're primarily targeting settings where optimizing C is prohibitively expensive, but I'd like to see some concrete pointers to papers in the "quickly growing body of work" you're referencing as a proof that such settings are indeed of interest to a wider audience.

---

> > > ### Author Response · Authors · 2019-11-15
> > > **Concrete references for applications**
> > >
> > > Dear reviewer,
> > >
> > > Thank you for your comment. We can imagine sum-and-sample with optimal C is hard to beat, but indeed we are mostly concerned with settings where this is not possible / prohibitively expensive due to the high-dimensional setting, such as in our VAE and TSP experiments. Please find below a number of additional references to papers that involve models with (mostly high-dimensional) discrete variables, many of them trained with reinforcement learning or Gumbel-Softmax.
> > >
> > > - Oord et al. 2017, Neural Discrete Representation Learning, NeurIPS
> > > - Kusner et al., 2017, Grammar Variational Autoencoder, ICML
> > > - Hu et al., 2017, Toward Controlled Generation of Text, ICML
> > > - Gu et al. 2018, Neural Machine Translation with Gumbel-Greedy Decoding, AAAI
> > > - Bahdanau et al. 2015, Neural Machine Translation by Jointly Learning to Align and Translate, ICLR
> > > - Ranzato et al., 2016, Sequence Level Training with Recurrent Neural Networks, ICLR
> > > - Rennie et al., 2017, Self-Critical Sequence Training for Image Captioning, CVPR
> > > - Yu et al., 2017, SeqGAN: Sequence Generative Adversarial Nets with Policy Gradient, AAAI
> > > - You et al., 2018, Graph Convolutional Policy Network for Goal-Directed Molecular Graph Generation, NeurIPS
> > > - Bunel et al., 2018, Leveraging Grammar and Reinforcement Learning for Neural Program Synthesis, ICLR
> > > - Balın et al., 2019, Concrete Autoencoders: Differentiable Feature Selection and Reconstruction, ICML
> > > - Kipf et al., 2018, Neural Relational Inference for Interacting Systems, ICML
> > > - Young et al., 2018, Recent trends in deep learning based natural language processing, IEEE Computational Intelligence Magazine

---

### Official Review · AnonReviewer2 · 2019-10-22
**Official Blind Review #2**

**Rating:** 6

**Review:**

Summary: In this paper, an unbiased estimator for expectations over discrete random variables is developed based on a sampling-without-replacement strategy. The proposed estimator is shown to be a Rao-Blackwellization of three existing unbiased estimators with guaranteed reduction in estimation variance. The connections of the method to other gradient estimators are discussed. Experimental results on several toy and real-data DL/RL problems are reported to demonstrate the applicability of the proposed estimators in the practice of machine learning.

Strong points:

-S1. The addressed topic is interesting and timely in deep/reinforcement learning.

-S2. The numerical results show some promise of the proposed estimator in reducing the gradient estimation variance in practice.

Weak points:

-W1. A formal and detailed problem statement is missing. The paper quickly jumps from a high-level problem setup description in the introduction section into some technical details in the preliminary & methodology sections, without any formal definition of the so called unordered set policy gradient estimation problem provided. What is the input/output of the estimator? Why this problem is important and challenging? It will be better to provide one or two concrete examples such as those studied in the experiments to give a more complete picture of the problem in study.

-W2. The motivation of study is not clearly elaborated. There are several existing options to reduce the gradient estimation variance via Rao-Blackwellization [see, e.g., Liu et al. 2019]. In which regimes the current method is more preferable than those prior ones and why? The justification of using the without-replacement-sampling strategy so far remains largely unconvincing.

-W3. The overall novelty of theory is limited. It makes sense that gradient estimators based a mini-batch of sample under without-replacement-sampling should tend to have smaller variance than the single-sample stochastic gradient estimation. Although a guarantee of variance reduction from the perspective of Rao-Blackwellization looks promising, the proof technique is fairly standard and more importantly, the quantification of such a variance reduction remains largely unaddressed. Therefore, the overall degree of novelty in theory is still relatively low.

-W4. The paper presentation quality can be improved. As another consequence of the above mentioned issues with problem statement and motivation, I found the paper a bit hard to follow smoothly. Particularly, too much space is spent on presenting the technical details while the principles/intuitions behind these fancy mathematical treatments are lacking in explanation. There are three theorems established in the paper, but none of them come up with sufficient discussions on the main messages conveyed by these results.

=== update after author response ===

Thank you for your response. I find my concerns on motivation and presentation properly addressed in the feedback and the revised paper as well.  Concerning the strength of theory, however, I am still not convinced that the current analysis is strong enough to thoroughly justify the benefit of the proposed estimator. All in all, the paper is substantially improved in presentation and the proposed gradient estimator seems to be a novel and more attractive alternative to the existing ones in a number of popular DL/RL applications. I thus would like to increase the rating to weak accept.


**Experience Assessment:**

I do not know much about this area.

**Review Assessment: Checking Correctness Of Derivations And Theory:**

I assessed the sensibility of the derivations and theory.

**Review Assessment: Checking Correctness Of Experiments:**

I assessed the sensibility of the experiments.

**Review Assessment: Thoroughness In Paper Reading:**

I read the paper at least twice and used my best judgement in assessing the paper.

---

> ### Author Response · Authors · 2019-11-12
> **Detailed response to Reviewer 2**
>
> Thank you for your time reviewing our paper and for the feedback! We have used it to update the paper. Please see our detailed reply below:
>
> - W1 problem statement
> The problem is gradient estimation of functions over distributions involving discrete variables. This is the same problem that is addressed by the popular Gumbel-Softmax estimator. Instead of being based on continuous relaxations, which is biased and does not work well in high-dimensional settings, we use the idea of using multiple samples. By sampling without replacement, we reduce variance compared to sampling with replacement. We have improved the introduction to give a better problem description and motivation.
>
> - W2 motivation
> The estimator by Liu et al. (2019) only works in low-entropy regimes when significant probability mass is concentrated on just a few categories. Our estimator has the same effect in the low-entropy regime, but is superior in the high entropy regime, in which case it ties with VIMCO which does not work well in the low-entropy regime (because of the with-replacement sampling strategy). In Figure 4, our estimator outperforms both VIMCO and Liu et al. (2019) (sum & sample).
>
> - W3 novelty of theory
> To the best of our knowledge, a practical, unbiased estimator under weighted sampling without replacement does not currently exist in the literature (the importance-weighted estimator has high variance). Our contribution is to derive such an estimator, and to connect it theoretically to three different estimators using Rao-Blackwellization, motivating its improvement. Additionally, we show how to include a built-in baseline while remaining unbiased, which is the key to making the estimator usable for gradient estimation (therefore an alternative to e.g. Gumbel-Softmax), but non-trivial since samples are dependent while baselines should be independent. We do show reduced variance in our experiments, although indeed it would be interesting to analyze the variance reduction analytically.
>
> - W4 paper presentation
> We may have compromised a bit on this aspect indeed. We have improved the readability of the paper by adding explanations on the implications of the theorems.

---

### Official Review · AnonReviewer1 · 2019-10-23
**Official Blind Review #1**

**Rating:** 6

**Review:**

Summary: This paper introduces an gradient estimator for loss functions that are expectations over discrete random variables. The basic idea is that an estimator over a discrete distribution can be Rao-Blackwellized by conditioning on the event that the discrete realization was produced by being the first sample drawn from an unordered set of samples drawn with replacement. Much of the paper is spent showing how this Rao-Blackwellized estimator can be computed in practice and how it compares to other known estimators.

Originality: This idea is original and quite nice.

Clarity: The paper is very easy to understand and well-written.

Quality:
- The derivations are all correct from what I can tell, and easy to read.
- The experiments are all reasonably well done, but I could not find where the authors report which optimizer was used and how the hyperparameters were set. If I missed it, could the authors point that out in the rebuttal? If it is missing, then I strongly suggest that the authors include that in the paper. Even if the training protocol is taken from another code base, I think this paper should be reasonably self-contained. How the hyperparameters are tuned and which optimizer was used could affect the interpretation of the results. Additionally, if overfitting is an issue, I recommend the authors consider the use of regularizers, like weight decay. I see no reason that this would violate the spirit of the paper, and might make their results more compelling.

Significance:
-This paper has some interesting ideas, and it is written well. So, it may inspire future work. Yet, from the experiments it is not obvious that this estimator adds much to the existing literature. In all experiments at least one known estimator matches the performance of the unordered set estimator.
- All of these estimators, which require multiple evaluations of the gradient, are generally less common in practice.

**Experience Assessment:**

I have published in this field for several years.

**Review Assessment: Checking Correctness Of Derivations And Theory:**

I carefully checked the derivations and theory.

**Review Assessment: Checking Correctness Of Experiments:**

I assessed the sensibility of the experiments.

**Review Assessment: Thoroughness In Paper Reading:**

I read the paper thoroughly.

---

> ### Author Response · Authors · 2019-11-12
> **Detailed response to Reviewer 1**
>
> Thank you for your time reviewing our paper and for the feedback! We have used it to update the paper. Please see our detailed reply below:
>
> - Details on hyperparameters
> Thank you for pointing this out, we underestimated the importance of including these details (since they are in the original papers). We agree that it is better to make the paper self-contained and we have added experiment details in the Appendix (with proper citation to the original experiments).
>
> - Regularization
> We considered the use of regularizers, but changing the experiment setup would make our results incomparable to the results by Yin et al. (2019). Our goal is to show the quality of our estimator, which is directly measured by gradient variance and indirectly by its success in optimizing the training performance. We agree that adding regularization would not violate the spirit of the paper, but we consider this orthogonal to the contribution in our paper (the same argument is made, e.g., by Grathwohl et al., 2018). Nevertheless, an alternative to using weight decay would be early-stopping, in which case our estimator achieves lowest validation performance the fastest, as can be seen in Figure 5.b) in the Appendix.
>
> - Performance matched by other estimators
> Yes, but the unordered set estimator is the only estimator that performs well across different settings (high and low entropy). Therefore it is more robust and a strict improvement to any of these estimators which only have good performance in either high or low entropy settings. Practically, this means our estimator removes the need to try different estimators that may work in different settings. Also, as typically the entropy shifts from high to low as the model gains confidence during training, our estimator allows to learn with maximal efficiency, both in early and late stages of training.
>
> - Multiple samples less common in practice
> This is true, but we think it should not be! The problem that we address (gradient estimation with discrete variables) is common in practice (given the popularity of Gumbel-Softmax), and we think the use of multiple samples as a solution is underappreciated. It is simple and effective and applicable to most (if not all) of the scenarios where Gumbel-Softmax is applied. In our experience, training using k > 1 samples (w or w/o replacement) for N/k epochs outperforms N epochs of Gumbel-Softmax/REBAR/RELAX, at the same computational cost, while being (apart from k) parameter free and unbiased, and allow to use a built-in baseline. In many cases, we think not considering multi-sample estimators ‘because they use multiple samples’ is not a valid reason.

---

### Author Response · Authors · 2019-11-12
**Updates to the paper based on feedback from reviewers**

We thank the reviewers for their feedback! We have used it to make a number of improvements to the paper. To summarize, roughly in order as in the paper, we have:
	• Improved the introduction to include a better problem description and motivation.
	• Added explanations of the implications of the theorems.
	• Improved the section on the sum-and-sample estimator, adding the discussion on optimally choosing the number of summed terms and why this is difficult in high-dimensional settings. Inspired by this we have generalized the results for the stochastic sum-and-sample estimator to include the setting with fewer summed terms and multiple samples.
	• Added a discussion on the relation to and limitations of stratified/systematic sampling.
	• Added  stratified and systematic sampling as baselines for the Bernoulli experiment (Figure 1).
	• Improved the caption for Figure 1 to explain that the sum-and-sample estimators with sampled baselines use more evaluations.
	• Added details about the experiments for VAE and TSP (in the Appendix).
	• Added explanation why we don't use regularization for the VAE experiment (since the goal is improved optimization and to keep results comparable to Yin et al. (2019)).
	• Moved Figure 2 to the Appendix and replaced it by a table, which makes the results more clear.
	• Added results for TSP with the normalized importance-weighted policy gradient estimator (with built-in baseline) based on an unpublished workshop paper of us (citation kept anonymous). This can be considered the priority sampling baseline by Duffield et al. (2007).
	• Added details on the numerical integration of p(S) in the Appendix. An implementation is available in the code.

---

### Decision · Program_Chairs · 2019-12-19

**Decision:**

Accept (Spotlight)

**Comment:**

The authors derive a novel, unbiased gradient estimator for discrete random variables based on sampling without replacement. They relate their estimator to existing multi-sample estimators and motivate why we would expect reduced variance. Finally, they evaluate their estimator across several tasks and show that is performs well in all of them.

The reviewers agree that the revised paper is well-written and well-executed. There was some concern about that effectiveness of the estimator, however, the authors clarified that "it is the only estimator that performs well across different settings (high and low entropy). Therefore it is more robust and a strict improvement to any of these estimators which only have good performance in either high or low entropy settings." Reviewer 2 was still not convinced about the strength of the analysis of the estimator, and this is indeed quantifying the variance reduction theoretically would be an improvement.

Overall, the paper is a nice addition to the set of tools for computing gradients of expectations of discrete random variables. I recommend acceptance.